



# McRALI: a Monte Carlo high spectral resolution lidar and Doppler radar simulator for three-dimensional cloudy atmosphere remote sensing

Frédéric Szczap[1], Alaa Alkasem[1], Guillaume Mioche[1,2], Valery Shcherbakov[1,2], Céline Cornet[3], Julien Delanoë[4], Yahya Gour[1,5], Olivier Jourdan[1], Sandra Banson[1], Edouard Bray[1]

[1]Université Clermont Auvergne, CNRS, UMR 6016, Laboratoire de Météorologie Physique (LaMP), 63178 Aubière, France
[2]Université Clermont Auvergne, Institut Universitaire de Technologie d'Allier, 03100 Montluçon, France
[3]Université Lille, CNRS, UMR 8518, Laboratoire d'Optique Atmosphérique (LOA), F-59000 Lille, France
[4]Université de Versailles Saint-Quentin-en-Yvelines, Université Paris-Saclay, Sorbonne Université, CNRS, Laboratoire Atmosphère, Milieu, Observations Spatiales (LATMOS), Institut Pierre Simon Laplace (IPSL), Guyancourt, France
[5]Université Clermont Auvergne, Institut Universitaire de Technologie d'Allier, 03200 Vichy, France

Correspondence to: Frédéric Szczap (szczap@opgc.univ-bpclermont.fr)

**Abstract.**

The aim of this paper is to present the Monte-Carlo code McRALI that provides simulations, under multiple scattering regimes of polarized high spectral resolution (HRS) lidar as well as Doppler radar observations for three-dimensional (3D) cloudy atmosphere. The effects of non-uniform beam filling (NUBF) on HSR lidar and Doppler radar signals related to the EarthCARE mission are investigated with the help of an academic 3D box-cloud, characterized by a single isolated jump in cloud optical depth, assuming vertically constant wind velocity. Regarding Doppler radar signals, it is confirmed that NUBF induces a severe bias in velocity estimates. The correlation of the NUBF bias of Doppler velocity with the horizontal gradient of reflectivity shows a correlation coefficient value around 0.15 m.s$^{-1}$(dBZ.km$^{-1}$)$^{-1}$ close to that given in scientific literature. Regarding HSR lidar signals, we confirm that multiple scattering processes are not negligible. We show that NUBF effects on molecular, particulate and total attenuated backscatter are mainly due to unresolved variability of cloud inside the receiver field of view, and to a lesser extent, to the horizontal photon transport. This finding gives some insight into the reliability of lidar signal modelling using independent column approximation (ICA).



## 1. Introduction

Spaceborne atmospheric LIDAR (Light Detection And Ranging) and RADAR (Radio Detection And Ranging) are suitable tools to investigate vertical properties of clouds on a global scale. Over the last decade, the Cloud-Aerosol Lidar and Infrared Pathfinder Satellite Observations (CALIPSO) (Winker et al., 2010) and the Cloud Satellite (CloudSat) (Stephens et al., 2008) improved our understanding of the spatial distribution of microphysical and optical properties of clouds and aerosols (Stephens et al., 2018). However, clouds remain the largest source of uncertainty in climate projections (Boucher et al., 2014, Dufresnes and Bony, 2008). Like clouds, aerosols are another large source of uncertainty in climate models (both direct and indirect radiative forcing) (see, e.g., Hilsenrath and Ward (2017) and references therein). CALIPSO and CloudSat missions were then extended for other 3 years (see, e.g., Vandemark et al., 2017). Future missions are planned to pursue those observations. For example, the Earth Clouds, Aerosol and Radiation Explorer (EarthCARE) (Illingworth et al., 2015) is scheduled for 2022, which will deploy for the first time in space the combination of a high resolution spectral (HSR) lidar and a Doppler radar. More recently, following the Atmospheric Dynamics Mission ADM-Aeolus (ESA report, 2016) by the European Space Agency (ESA), an atmospheric dynamics observation satellite was placed in orbit on August 2018, which deployed the first space Doppler lidar. The Atmospheric LAser Doppler INstrument (ALADIN) of the ADM-Aeolus, the ATmospheric LIDar (ATLID) and the Cloud Profiling Radar (CPR) of the EarthCARE mission will provide spectrally resolved data. The CPR will allow to account for the Doppler effect and will provide information on convective motions, wind profiles and fall speeds (Illingworth et al., 2015). The ATLID will perform measurements of extinction coefficient and lidar ratio (ESA, 2016; Illingworth et al., 2015).

Simulation tools are steadily advancing hence allowing to explore direct and inverse problems in a cost-effective way. And, simulators of lidar and/or radar signals are no exception. In this introduction, published works restricted to the case when multiple scattering was taken into account are briefly discussed. Fruitful findings, mostly on lidar returns from clouds, were obtained by the MUSCLE (MUltiple SCattering in Lidar Experiments) community in the nineties. A review of the participating models can be found in the work by Bissonnette et al. (1995). A Monte Carlo (MC) model was used by Miller and Stephens (1999) to study the specific roles of cloud optical properties and instrument geometries in determining the magnitude of lidar pulse stretching. Several models, which take into consideration Stokes parameters, were developed in the 2000s (Hu et al., 2001; Noel et al., 2002; Ishimoto and Masuda, 2002; Battaglia et al., 2006). Fast approximate lidar and radar multiple-scattering models (Chaikovskaya, 2008; Hogan, 2008; Hogan and Battaglia, 2008; Sato et al., 2019) provide possibility, for example, to explain certain important characteristics of the dual-wavelength reflectivity profiles (Battaglia et al., 2015), although the codes are inherently one dimensional. In addition, a comprehensive review of multiple-scattering in radar systems can be found in the work by Battaglia et al. (2010). The basic principles of Monte-Carlo models, which consider the Doppler effect and spectral properties of received signals,



were developed in the nineties for the needs of laser Doppler flowmetry (see, e.g., de Mul et al. (1995) and references therein). As for lidar and radar measurements, we can refer to the EarthCARE simulator (ECSIM) that is a modular multi-sensor simulation framework, where a fully 3D Monte Carlo forward model can calculate the spectral-polarization state of ATLID lidar signals (Donovan et al., 2008; Donovan et al., 2015). A radar DOppler MUltiple Scattering (DOMUS)

simulator can be run in a full 3D configuration and allows a comprehensive treatment of non-uniform-beam-filling (NUBF) scenarios (Battaglia and Tanelli, 2011).

The McRALI simulators (Monte Carlo modeling of RAdar and LIdar signals) developed at the Laboratoire de Météorologie Physique (LaMP) are based on 3DMcPOLID (3D Monte Carlo simulator of POLarized LIDar signals), a

MC code dedicated to simulate polarized active sensor signals from atmospheric compounds in single and/or multiple-scattering conditions (Alkasem et al., 2017). As their core they use the three-dimensional polarized Monte-Carlo atmospheric radiative transfer model (3DMCPOL, Cornet et al. (2010). As 3DMCPOL, they use the local estimate method (Marchuk et al., 1980; Evans and Marshak, 2005) to reduce the noise level and take into account the polarization state of the light. Photons are followed step by step through the cloudy atmosphere. At each interaction, the contribution to the

detector is computed according to the scattering matrix and the field of view (FOV) of the detector. Variance reduction techniques proposed by Buras and Mayer (2011) can be employed for the purpose of reducing noise due to the strong forward scattering peak and consequently increasing the computational efficiency.

The objective of this work is to describe the latest evolution of McRALI which provides the means to simulate High-

Spectral-Resolution (HSR) lidar and Doppler radar signals. The organization of this paper is as follows. In Sect. 2, we explain in detail the methodology used in McRALI to model spectral properties of lidar or radar data. Two illustrative applications (i.e. ATLID lidar and CPR radar of EarthCARE mission) of the developed simulator are presented. In Sect. 3 we briefly investigate errors induced by NUBF on the EarthCARE lidar and radar measurements with the help of the academic 3D box-cloud. This work is unique in that the results can be obtained only if the simulator is a fully 3D Monte

Carlo forward model. Conclusions and discussions are presented in Sect. 4.

## 2. Modelling of HSR lidar and Doppler radar signals with McRALI

### 2.1. General principles for the computation of frequency resolved signal

Basic lidar or radar equation can be written as (Weitkamp, 2005; Battaglia et al., 2010) :

$$p(r) = \frac{K(r)}{r^2}\beta(r)\exp\left[-2\int_0^r \sigma_{ext}(r')dr'\right]$$

(1)





where $p$ is the power on the detector from range $r$, $K$ is the instrument function, $\sigma_{ext}$ (in m$^{-1}$) is the extinction, and $\beta$ (in m$^{-1}$ sr$^{-1}$) is the backscattering coefficient defined as

$$\beta = P(\pi)\sigma_s \tag{2}$$

where $P(\pi)$ (in sr$^{-1}$) is the scattering phase function in the backward direction and $\sigma_s$ (in m$^{-1}$) is the scattering coefficient. Whereas the lidar community use the backscattering coefficient $\beta$, the radar community prefers to use the reflectivity $Z$ related

to $\beta$ as

$$Z = \frac{4}{|K_w|^2}\left(\frac{\lambda}{\pi}\right)^4 \beta \tag{3}$$

where $|K_w|^2$ is a dielectric factor usually assumed for liquid water and $\lambda$ is the wavelength. The radar reflectivity factor is expressed in mm$^6$.m$^{-3}$. However, due to its large dynamic range, it is more commonly expressed in dBZ and $10\log_{10}(Z)$ is used. Note that the reflectivity given in Eq. (3) is the radar non-attenuated reflectivity.

If the extinction value of the medium tends to zero, the measured backscatter $\hat{\beta}$ (or in the case of radar, the measured reflectivity

$\hat{Z}$) is equal to the "true" backscatter of the medium $\beta$(or $Z$) (Hogan, 2008).Under single scattering regimes, in an optically thicker medium, $\hat{\beta}(r) = \beta(r)\exp\left[-2\int_0^r \sigma_{ext}(r')dr'\right]$, and $\hat{Z}(r) = 4\lambda^4\hat{\beta}(r)/\pi^4|K_w|^2$. $\hat{\beta}(r)$ is then also called attenuated backscattering coefficient, hereafter also noted ATB. $\hat{Z}(r)$ is then the attenuated reflectivity. Under multiple scattering regimes, there is no rigorous analytical solution of $\hat{\beta}$ (or $\hat{Z}$). The common feature of the McRALI codes is that they provide range-resolved profiles of Stokes parameters $\mathbf{S}(r) = [I(r), Q(r), U(r), V(r)]$ and account for emitter and receiver shape

Generally speaking, high-spectral-resolution lidars of any type as well as Doppler radars share basic principles, that is the useful retrieved data are based on the spectral dependence of the recorded signals. Consequently, the Monte Carlo forward simulator has to account for an additional parameter, namely, the frequency shift when a photon interacts with a particle or the molecular atmosphere. The photon frequency has to be tracked through all scattering events until the photon is recorded by a receiver. Of course, it is computationally expensive to store the frequency value of all received photons. A solution developed

for the needs of Laser Doppler flowmetry (see, e.g., de Mul et al., 1995) was used by Battaglia and Tanelli (2011) in their DOMUS simulator. It consists to create discrete frequency distribution, which represents the number of photons with a Doppler shift in a certain frequency range. We follow that approach in the latest version of McRALI as our simulators provide Stokes parameters $\mathbf{S}(r,f) = [I(r,f), Q(r,f), U(r,f), V(r,f)]$, tabulated by range $r$ and frequency $f$ at the same time. $\mathbf{S}(r,f)$ is hereinafter referenced to "idealized polarized backscattered power spectrum profiles" or simply "power spectra". In other

words, the result of simulations is a two-dimensional matrix for each of the computed Stokes parameters, without considering

the Doppler spectrum folding depending on the measurement technology (step 2 in Fig.1). The generic name McRALI-FR will

be used for our Frequency-Resolved simulators.

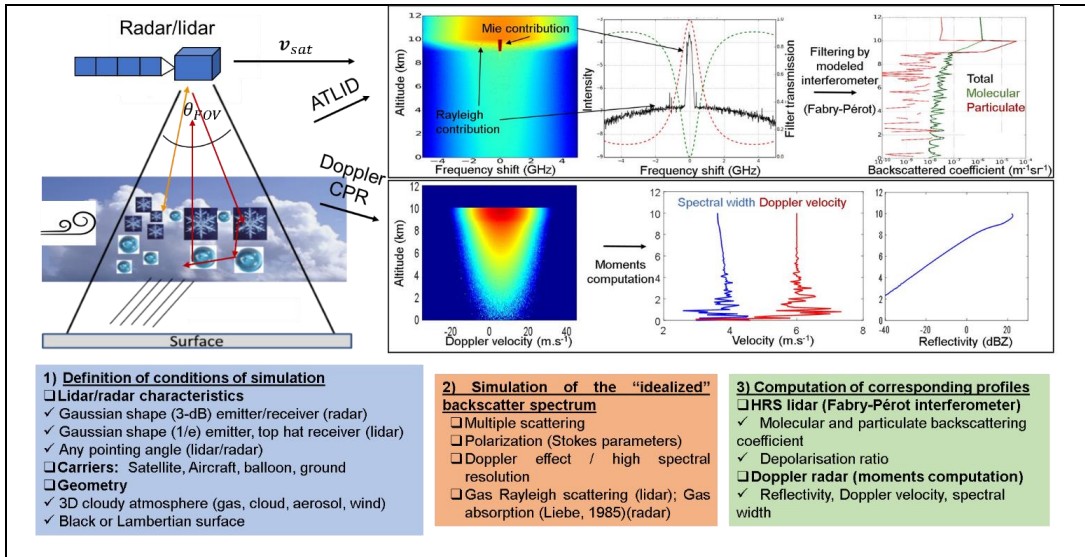

Figure 1: Schematic presentation of McRALI-FR simulator. Once the simulation conditions are defined (step 1), McRALI

calculates the idealized backscatter spectrum (step 2). In the last step (step 3), using a dedicated software, the desired

quantity profiles are calculated. Note that cloud extinction between 9 and 10 km altitude is set to 3 km$^{-1}$ for both lidar and

radar simulation.

The simulation conditions (step 1 in Fig.1) consists to set the 3D optical and dynamical properties of the cloudy atmosphere,

the surface and main characteristics of the instrument (currently monostatic high-spectral-resolution lidar or a Doppler radar),

that are its spatial position, its velocity, the viewing direction, the frequency and the polarization state of the emitted radiation,

and the shape of the emitter as well as the receiver. If at least one of those parameters varies, the simulation has to be carried

out once more even when 3-D cloudy atmosphere properties remain unchanged. Computations are carried out and profiles of

$\mathbf{S}(r, f)$ are stored in output files (step 2 in Fig.1). A separated software uses the saved files to account for spectral and

polarization characteristics of receivers and computes profiles of corresponding HSR lidar or Doppler radar signals (step 3 in

Fig.1), such as the particulate and molecular backscattering coefficient profiles for HSR lidar or reflectivity and Doppler

velocity profiles for Doppler radar.

The next five subsections describe in detail how McRALI-FR accounts for the Doppler effect, the modelling of transmitter

and receiver patterns, the Lambertian ground surface and present two examples of  McRALI-FR configuration in order to

simulate the HRS ATLID lidar and the Doppler CPR radar of the EarthCARE mission.





## 2.2. Modelling of idealized polarized backscattered power spectrum profiles

McRALI-FR accounts for phenomena that lead to the frequency shift of the received photon. This is the Doppler effect, which is due to the motion of gas (negligible for radar application), aerosol (negligible for radar application) and cloud particles. We use the term "cloud particles" for precipitating hydrometeors as well.

When both the source and the receiver are moving, the Doppler effect can be expressed in a ground-based frame of reference as follows (see, e.g., Tipler and Mosca, 2008):

$$f_r = f_s \left( \frac{1 - \frac{1}{c}\, \boldsymbol{v}_r . \boldsymbol{k}_{s,r}}{1 - \frac{1}{c}\, \boldsymbol{v}_s . \boldsymbol{k}_{s,r}} \right) \tag{4}$$

where $f_s$ and $f_r$ denote the frequencies; $\boldsymbol{v}_s$ and $\boldsymbol{v}_r$ are the velocity vectors; the source and receiver parameters are identified by the subscripts $s$ and $r$, respectively; $\boldsymbol{k}_{s,r}$ is the unit vector directed from the source to the receiver; $c$ is the speed of electromagnetic waves; $\boldsymbol{a} \cdot \boldsymbol{b}$ denotes the scalar product. If absolute values $|\boldsymbol{v}_s|$ and $|\boldsymbol{v}_r|$ of the velocities are both small compared to

the speed $c$, the series expansion of Eq. (4) takes the form:

$$f_r = f_s \left[ 1 - \frac{1}{c} (\boldsymbol{v}_r - \boldsymbol{v}_s) \cdot \boldsymbol{k}_{s,r} \right] \tag{5}$$

where the terms of the second order or higher than $1/c$ are neglected.

In the multiple scattering conditions, Eq. (5) can be rewritten for the scattering order $i$ as follows:

$$f_{i+1} = f_i \left[ 1 - \frac{1}{c} (\boldsymbol{v}_{i+1} - \boldsymbol{v}_i) \cdot \boldsymbol{k}_{i,i+1} \right], i = 0, 1, \dots, n \tag{6}$$

where $n$ is the total number of the scattering orders. The frequency $f_0$ of an emitted photon and the vector $\boldsymbol{v_0} = \boldsymbol{v_{n+1}} = \boldsymbol{v_{sat}}$ of the satellite velocity belong to the set of the input parameters of McRALI-FR.

In general, if a photon was scattered by particles $n$ times, its frequency at the lidar/radar receiver is expressed as follows:

$$f_n = f_0 \left[ 1 - \frac{1}{c} \sum_{i=0}^{n} (\boldsymbol{v}_{i+1} - \boldsymbol{v}_i) \cdot \boldsymbol{k}_{i,i+1} \right] \tag{7}$$

All terms of the second order or higher than $1/c$ are neglected as above. The unit vector $\boldsymbol{k_{0,1}}$ is directed from the satellite to the first scatterer; $\boldsymbol{k_{n,n+1}}$ is directed from the last scatterer to the satellite. It should be noted that Eq. (7) is in agreement with Eq. (5) of the work by Battaglia and Tanelli (2011), where, at the scattering order $i$, the frequency shift $\Delta f_i$ can also be given by





$$\Delta f_i = \frac{f_0}{c} \boldsymbol{v_i}.\left(\boldsymbol{k_{i-1,i}} - \boldsymbol{k_{i,i+1}}\right) = \frac{f_0}{c} \boldsymbol{v_i}.\left(\boldsymbol{k_{i-1}} - \boldsymbol{k_i}\right) \tag{8}$$

Figure 2 shows a schematic diagram of the frequency-shift consideration, at each interaction, by using the local estimate method. A photon path of two scattering events within the lidar/radar FOV is represented in red. Velocity of the first and the second scatterer is $\boldsymbol{v_1}$ and $\boldsymbol{v_2}$, respectively. At the first and second scattering, frequency shift is $\Delta f_1 = \frac{f_0}{c} \boldsymbol{v_1}.(\boldsymbol{k_0} - \boldsymbol{k_1})$ and

$\Delta f_2 = \frac{f_0}{c} \boldsymbol{v_2}.(\boldsymbol{k_1} - \boldsymbol{k_2})$, respectively. At each scattering event, McRALI-FR uses the local estimate method to compute the

5    contribution to the detector. For example, at the second scattering event, the total frequency shift is computed as $\Delta f_{2;total} =$

$\Delta f_1 + \Delta f'_2$, where $\Delta f'_2 = \frac{f_0}{c} \boldsymbol{v_2}.(\boldsymbol{k_1} - \boldsymbol{k'_2})$, $\boldsymbol{k'_2}$ being the direction from the second scattering event to the detector (dotted blue line) which works with the local estimate method. The frequency shift due to satellite motion is deliberately ignored to simplify the scheme but it is present in the codes. Computation of McRALI-FR power spectrum can also be performed following the convention of the "Gaussian approach" proposed by Battaglia and Tanelli (2011).

10   Note that in the current version of McRALI-FR codes, the wind velocity can be set by the user or provided by Large Eddy Simulation models at their grid scale. Sub-grid turbulence wind velocity is supposed to be homogeneous and isotropic; the turbulence velocity vector $\boldsymbol{v_{turb}}$ is distributed according to a Gaussian probability density function (PDF) (see, Wilczek et al. (2011) and references therein). The single-point velocity PDF has zero mean and the standard deviation $\sigma_{turb}$ for all three coordinates of $\boldsymbol{v_{turb}}$. The multivariate normal distribution is generated using the Box–Muller method (see, e.g., Tong (1990)).

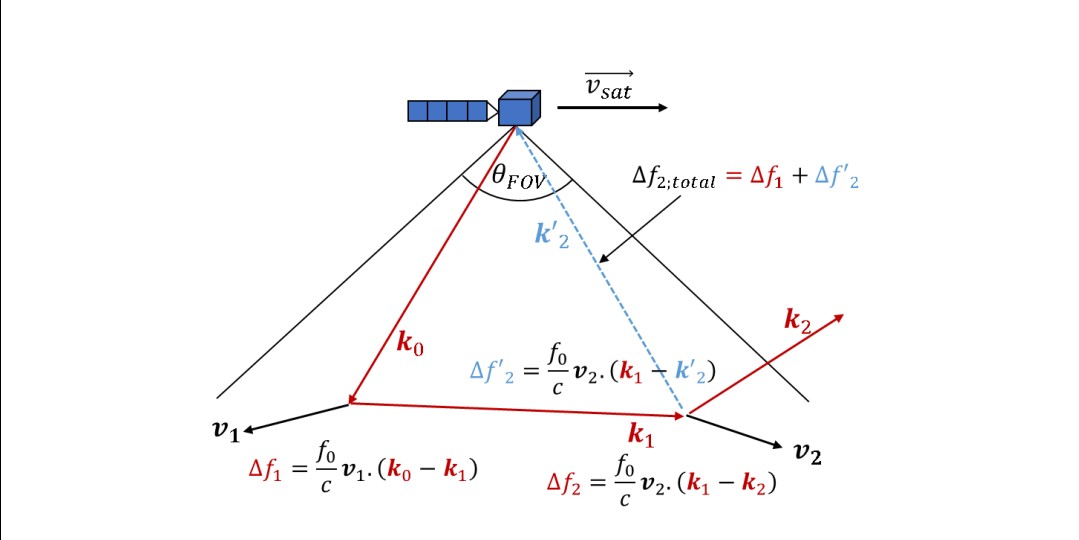

Figure 2: Schematic diagram of frequency-shift consideration along the propagation of photons in scattering medium in the framework of the locate estimate method.





## 2.2. Modelling of transmitter and receiver pattern

The current version of McRALI-FR codes only allows the monostatic configuration of transmitters and receivers of lidar or radar systems. Lidar/radar systems can be positioned at any altitude, allowing for ground-based, spaceborne and airborne configurations, with any viewing direction. Lidar transmitter is assumed to be a Gaussian laser beam with $1/e$ angular half-

width $\theta_{laser}$. For instance, a Gaussian laser beam pattern with $1/e$ angular half-width $\theta_{laser}$ is described by (Hogan, 2008)

$$g_1(\theta) = \exp\left[-\left(\frac{\theta^2}{\theta_{laser}^2}\right)\right] \tag{9}$$

Lidar receiver is assumed to be a top-hat telescope with a half-angle field of view $\theta_{FOV}$ and its pattern can be described by (Hogan and Battaglia, 2008)

$$g_2(\theta) = \begin{cases} 1 \; ; \; \theta \leq \theta_{FOV} \\ 0 \; ; \; \theta > \theta_{FOV} \end{cases} \tag{10}$$

Radar transmitters and receivers are assumed to be Gaussian antennas with a 3-dB half-width $\theta_{FOV}$. For instance a Gaussian antenna pattern with 3-dB half- width $\theta_{FOV}$ is described by (Battaglia et al., 2010)

$$g_3(\theta) = \exp\left[-\ln2\left(\frac{\theta^2}{\theta_{FOV}^2}\right)\right] \tag{11}$$

The lidar and radar transmitter and receiver pointing direction is defined by the zenith $\Theta_0$ and azimuthal $\phi_0$ angles. Direction cosines $(u_0, v_0, w_0)$ of the initial photon leaving the transmitter, calculated in the same way as Battaglia et al. (2006), are given by:

$$u_0 = a_1 \cos\Theta_0 \cos\phi_0 - a_2 \sin\phi_0 + a_3 \sin\Theta_0 \cos\phi_0 \tag{12.1}$$

$$v_0 = a_1 \cos\Theta_0 \sin\phi_0 + a_2 \cos\phi_0 + a_3 \sin\Theta_0 \sin\phi_0 \tag{12.2}$$

$$w_0 = -a_1 \sin\Theta_0 + a_3 \cos\Theta_0 \tag{12.3}$$

where $a_1 = x_1/(1 + x_1^2 + x_2^2)^{1/2}$, $a_2 = x_2/(1 + x_1^2 + x_2^2)^{1/2}$, $a_3 = 1/(1 + x_1^2 + x_2^2)^{1/2}$ with $x_1 = \tan\eta$ and $x_2 = \tan\xi$. To reproduce the Gaussian pattern of Eq.(9) and Eq.(11), $\eta$ and $\xi$ are Gaussian-distributed random numbers with zero mean

and standard deviation equal to $\theta_{laser}/\sqrt{2}$ and $\theta_{FOV}/\sqrt{2\ln2}$, respectively. The multivariate normal distribution is generated using the Box–Muller method (see, e.g., Tong (1990)).

## 2.3. Modelling of a Lambertian surface

The current version of McRALI-FR code uses the Lambertian-surface model. The probability that a photon is scattered by the surface is defined by the albedo $\Lambda$. When $\Lambda = 0$, i.e., the black surface model, it is assumed that all photons are absorbed by



the surface. Otherwise, i.e., $0 < \Lambda \leq 1$, the photon weight is multiplied by $\Lambda$. All photons scattered by the Lambertian surface are depolarized, i.e., have the Stokes parameters of the form $\mathbf{S} = [I, 0, 0, 0]$. The interaction of a photon with the surface is treated in the same way as the scattering by a cloud or aerosol particle or the Rayleigh scattering (Cornet et al., 2010).

First, the new direction of a photon scattered by the surface is random and it is simulated according to the well-known algorithm

5   (see, e.g., Mayer, 2009). The azimuth angle $\varphi$ is chosen randomly between 0 and $2\pi$:

$$\varphi = 2\pi q_1 \tag{13}$$

as for the zenith angle $\theta$, its cosine $\mu = \cos(\theta)$ is randomly drawn using the expression:

$$\mu = -\sqrt{q_2} \tag{14}$$

where $q_1$ and $q_2$ are uniform random numbers between 0 and 1.

Secondly, the local estimate technique (Marchuk et al., 1980) is implemented to calculate at each scattering point the contribution of the photon in the direction of the sensor.

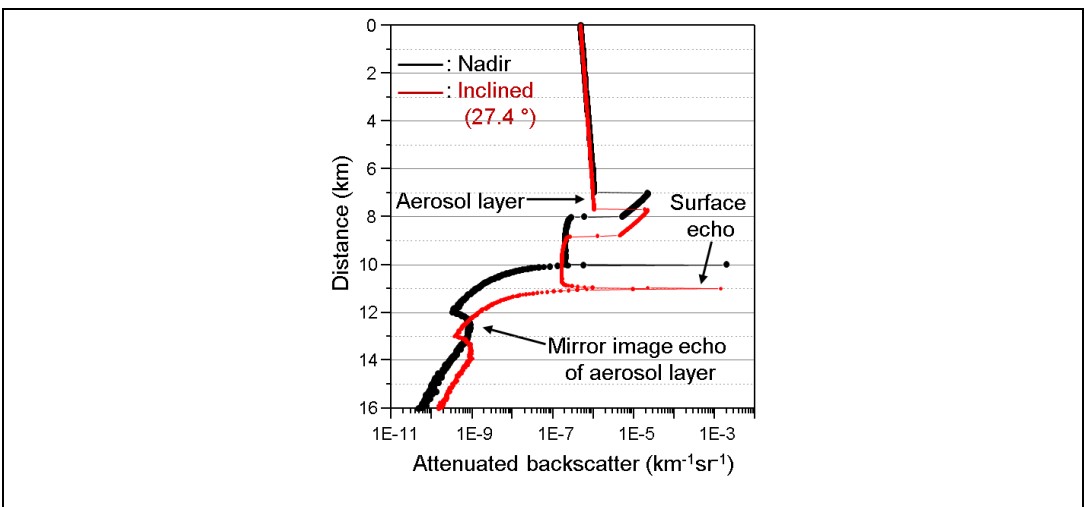

Figure 3: Profiles of the attenuated backscatter (ATB) coefficient (black – nadir looking, red – inclined at 24.7°) as a function of the distance from the lidar position. The lidar altitude is 10 km.

10   Figure 3 shows as an example the two lidar signals as a function of the distance from the lidar position for two viewing directions (nadir and inclined at 24.7 degrees, chosen so that the distance to the ground is 11 km). The lidar altitude is 10 km, the laser divergence is 0.0007 and the field of view of the receiver is 0.005 radians. An aerosol layer, between altitudes of 2 and 3 km, has an optical thickness of 0.15. The single scattering albedo of 0.91888 and the phase function were computed with





the refractive index and microphysical parameters of the coarse mode of desert dust and assuming that particles are spheroids with a distribution of the aspect ratio (Dubovik et al., 2006). The albedo of the Lambertian surface is set to 1.

For the nadir-direction example, the layer at distances between 7 and 8 km that exhibit large values of backscatter coefficient corresponds to the aerosol layer between 2 and 3 km in altitude. At a distance of 10 km, the very large value of the backscatter

coefficient corresponds to the echo from the surface. Then, for distances larger than 10 km, the lidar signal drastically decreases. But for the distances from 12 km to 13 km, another layer can be observed. That layer corresponds to a third and higher order of scattering. In this particular case, the triple scattering is of the type "surface – aerosol layer – surface". It is also called the mirror image and refers to reflectivities measured by airborne or spaceborne radars at ranges beyond the range of the surface reflection (see, e.g., Battaglia et al., 2010). It should be underscored that the mirror image disappears when, during

a simulation, one photon can undergo no more than two scatterings. The same behaviour is observed for the case of the inclined viewing direction. The position of the aerosol layer, the surface echo and the mirror image shift in agreement with corresponding distances from the lidar.

### 2.4. Doppler radar CPR/EarthCARE configuration

#### 2.4.1. Modelling gas absorption

At 94 GHz (3.2 mm, W-band), the attenuation by atmospheric gas is mainly due to absorption of water vapour and oxygen (Liebe, 1985; Lenoble, 1993; Liou, 2002). Attenuation $A$ (in dB.km$^{-1}$) by water vapour and oxygen in McRALI codes is computed from Liebe (1985) tabulations. Absorption coefficient $\sigma_{abs}$ (in km$^{-1}$) is given by $\sigma_{abs} = 0.2303\,A$. Absorption and scattering are treated separately in McRALI codes, as is done in 3DMCPOL (Fauchez et al., 2014), where absorption is considered by a photon weight $w_{abs}$ according to the Lambert-Beer's law ( Partain et al., 2000 ; Emde et al., 2011):

$$w_{abs} = e^{-\int_0^s \sigma_{abs}(s')ds'} \tag{15}$$

where $ds'$ is a path element of the photon path.

#### 2.4.2. Doppler spectrum and its relation to reflectivity, Doppler velocity and spectral width

The Doppler Radar community uses the Doppler spectrum $S(r, v)$, a power-weighted distribution of the radial velocities $v$ in the velocity range $dv$ of the scatterers (Doviak and Zrnić, 1984). McRALI-FR codes dedicated to Doppler radar simulations compute $S(r, v)$ by using the first Stokes parameter $I(r, f)$ and the Doppler formula $v = c\,f/2f_0$. We follow the convention

that the Doppler velocity is positive for motion away from the radar. The backscattering coefficient profile $\beta(r)$ is then given by:

$$\beta(r) = \int_{-\infty}^{+\infty} I(r, v)\, dv \tag{16}$$





The reflectivity $Z(r)$ profile is computed using Eq. (3) and $\beta(r)$. The Doppler velocity profile $V_{Dop}(r)$ is defined as

$$V_{Dop}(r) = \frac{\int_{-\infty}^{+\infty} v I(r,v)\,dv}{\int_{-\infty}^{+\infty} I(r,v)\,dv} \tag{17}$$

and the Doppler velocity spectral width profile $\sigma_{Dop}(r)$ is obtained from:

$$\sigma_{Dop}^2(r) = \frac{\int_{-\infty}^{+\infty} [v - V_{Dop}(r)]^2 I(r,v)\,dv}{\int_{-\infty}^{+\infty} I(r,v)\,dv} \tag{18}$$

Figure 4 shows, as an example, a simulation of the Doppler power spectrum, the Doppler velocity, the Doppler velocity spectral width and the reflectivity profiles for a CPR/EarthCARE-like radar for a homogenous iced cloud layer with fixed 6 m.s$^{-1}$

downdraft at all altitudes (see details of conditions of simulation in Tab. 1) with ($\sigma_{turb} = 0.5$ ms$^{-1}$) and without ($\sigma_{turb} = 0$ ms$^{-1}$) sub-grid turbulent wind. In a first step, McRALI-FR codes dedicated to Doppler radar simulations compute the idealized Doppler power spectrum density $\mathbf{S}(r,v)$. The first Stokes parameter $I(r,v)$ of the Doppler spectrums (with and without sub-grid turbulent wind) are shown in Fig. 4a and Fig. 4b, respectively. Then, in a second step, a software computes the reflectivity, the Doppler velocity and the Doppler velocity spectral width profiles with Eq.16, Eq.17 and Eq.18, respectively. Multiple

scattering (MS) and single scattering (SS) Doppler velocity profiles are superimposed on the MS Doppler spectrum. MS and SS Doppler velocity values are constant within the cloud layer (between 9 km and 10 km of altitude) and are equal to the "true" 6 ms$^{-1}$ vertical velocity, whatever the wind turbulence value is. Due to multiple scattering processes, the apparent Doppler velocity of 6 ms$^{-1}$ can be observed between the cloud base altitude and the ground, contrary to the SS apparent Doppler velocity which appears only in the cloud layer.

On Fig. 4c the MS and SS Doppler velocity spectral width profiles are drawn. Under SS approximation, the Doppler velocity spectral width is given by (Tanelli et al., 2002):

$$\sigma_{Dop}^2 = \sigma_{turb}^2 + \left( \frac{\rho_R v_{sat}}{2\sqrt{\ln(2)}} \right)^2 \tag{19}$$

where $v_{sat}$ is the satellite velocity relative to the ground and $\rho_R$ is the Gaussian (3-dB) FOV half-angle. Simulated SS Doppler velocity spectral width without turbulence and with turbulence are close to 3.58 m.s$^{-1}$ and 3.62 m.s$^{-1}$, respectively. Both computed values are very close to theory predicted values. On the other hand, MS processes together with sub-grid turbulent

wind are a source of broadening. For example, at 2 km under the cloud base, $\sigma_{Dop} = 3.75$ m.s$^{-1}$, which is larger than the SS value.





Vertical profiles of MS and SS reflectivity are shown on Fig. 4d. These profiles are not sensitive to the wind turbulence. MS processes are a source of enhancement of the reflectivity compared to the SS reflectivity and the apparent reflectivity that can be observed under the cloud layer.

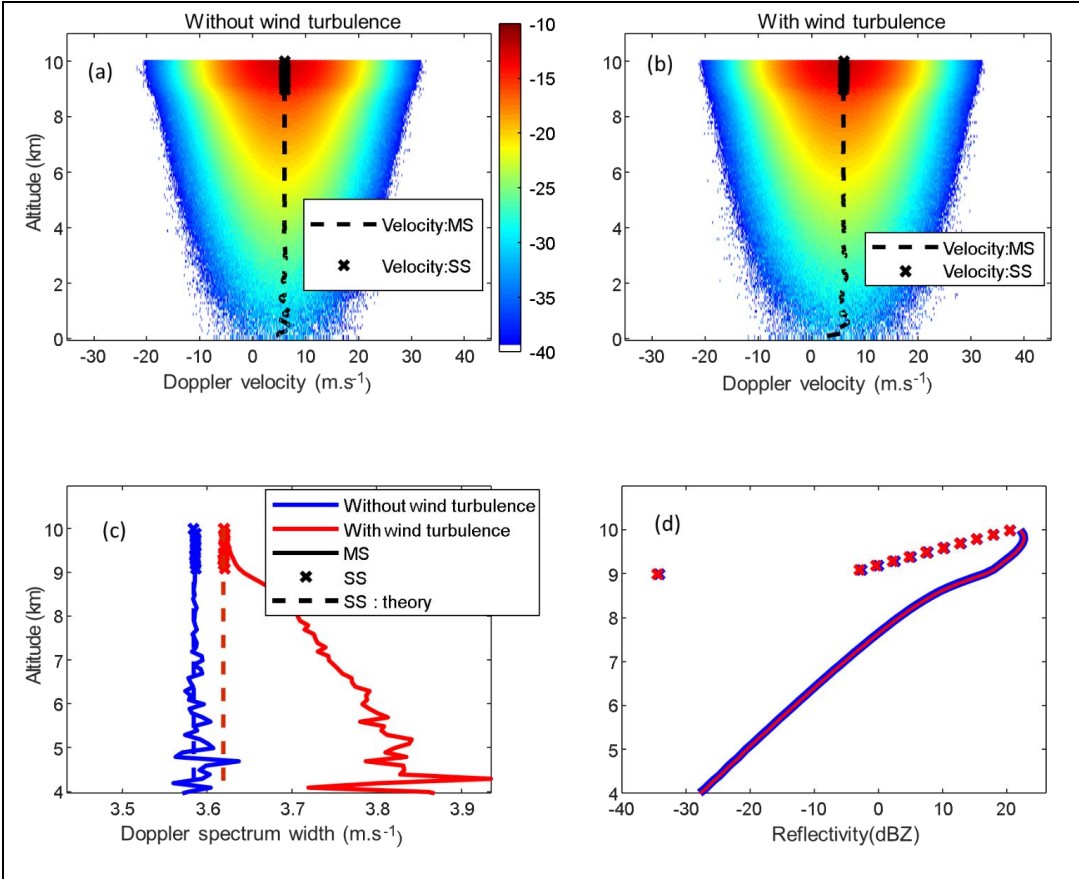

Figure 4: Estimated Doppler spectrum moments for Doppler CPR/EarthCARE-like radar. (a) Doppler spectrum without wind turbulence. Doppler velocity profiles are superimposed (MS: dotted line, SS: cross). (b) Same as (a), but with wind turbulence. (c) Vertical profiles of MS (full lines) and SS (crosses) Doppler spectrum width with wind turbulence (red) and without wind turbulence (blue). Dotted lines indicate predicted values by theory. (d) Vertical profiles of MS (full lines) and SS (crosses) reflectivity with wind turbulence (red) and without wind turbulence (blue). The altitude of the base of the iced homogeneous cloud layer (optical depth of 3) is 9 km. Its geometrical thickness is 1 km.


### 2.5. High spectral resolution (HRS) lidar ATLID/EarthCARE configuration

#### 2.5.1. Modelling of the emitted laser energy spectrum

The laser transmitter of the ATLID instrument has spectral requirements with a spectral linewidth below 50 MHz (Hélière et al., 2017). In McRALI-FR codes, frequency of the emitted radiation is drawn randomly according to a Gaussian law of average

$f_0$ with a $1/e$ half-width $\sigma_{f_0} = 50$ MHz.

#### 2.5.2. Modelling of thermal molecular velocity distribution

The current version of McRALI-FR codes assumes that each component of molecular velocity is distributed according to the Maxwell-Boltzmann density function with null mean and standard deviation $a$ given by:

$$a = \sqrt{\frac{kT}{m}} \tag{20}$$

where $k$ is the Boltzmann's constant, $T$ is the temperature and $m$ is the molecular mass of gas The multivariate normal

distribution is generated using the Box-Muller method. As a next step, we plan to take into account spontaneous Rayleigh-Brillouin scattering.

#### 2.5.3. Relation of the HSR spectrum to molecular and particulate backscattering coefficient: modelling of a Fabry-Pérot interferometer

One of the important features of HSR lidars is the possibility to retrieve profiles of particle extinction and backscattering

coefficient without the need for additional information on the lidar ratio (Shipley et al., 1983; Ansmann et al., 2007 and references therein). HSR technology relies on the principle of measuring Doppler frequency shift resulting from the scattering of photons by molecules (referred as molecular scattering or Rayleigh scattering) and by particles (referred as particulate scattering or Mie scattering). The characteristic shape of HSR spectrum depends on both these two scattering processes: broad spectrum of low intensity for molecules scattering and narrow peak of large intensity for the particles scattering.

The spectral width of the particles peak will be determined by the spectral width of the laser pulse itself along with any turbulence present in the sampling volume. The spectral width of the ATLID laser will be on the order of 50 MHz so that the laser line width would be the dominant factor. Thus, the molecular backscatter will be much broader than the particulate scattering return. This is due to the fact that the atmospheric molecules have a large thermal velocity. Assuming Gaussian molecular thermal velocity distribution with a half-width at $1/e$ of the maximum, molecular broadening $\gamma_m$ can be written

(Bruneau and Pelon, 2003):

$$\gamma_m = \frac{2}{\lambda_0} \sqrt{\frac{2kT}{m}} \tag{21}$$





If $T = 230$ K, then $\gamma_m$ is in the order of 2 GHz, which is about 40 times larger than the laser line width. Thus, using interferometers (such as a Fabry–Pérot interferometer (FP) equipping the ATLID/EarthCARE lidar) and appropriate signal processing (Hélière et al., 2017), the molecular and particulate contributions of the lidar backscattered signal can be separated. Then particulate and molecular backscattering coefficients profiles (attenuated or apparent attenuated backscattering

coefficient or simply attenuated backscatter, also noted ATB) can be separately determined. In this study, we suppose that the FP interferometer has the following parameters. The free spectral range is 7.5 GHz, the finesse is 10, and the FP is centered at the wavelength 355 nm. The cross-talk effects were taken into account according to the work by Shipley et al. (1983). The coefficients of the cross-talk correction were computed using an Airy function (see, e.g., Vallée and Soares, 2004), which describes the FP transmission spectrum, assuming a Gaussian molecular thermal velocity distribution with a half-width at $1/e$

of the maximum $\gamma_m$ (Eq. 21). This method determines four calibration coefficients corresponding to the fraction of cloud/aerosol backscatter in the molecular and particulate channels ($C_{am}$ and $C_{aa}$, respectively), and the fraction of molecular backscatter in the molecular and particulate channels ($C_{mm}$ and $C_{ma}$, respectively). The calculation method of these coefficients is described in detail in Shipley et al. (1983). As an indication, for the present study in the ATLID/EarthCARE lidar configuration, these coefficients have the following values: $C_{mm} = 0.543$, $C_{ma} = 0.457$, $C_{aa} = 0.998$ and $C_{am} =$

$0.002$.

Figure 5 shows particulate and molecular ATB profiles for ATILD/EarthCARE-like lidar. We consider in this example an ice cloud, corresponding to a homogenous layer with optical depth of 3 between 9 and 10 km in altitude (see details of simulation conditions in Sect. 3.1). In the first step, McRALI-FR codes, dedicated to HRS lidar simulations, compute the HRS spectrum $\mathbf{S}(r, f)$. The first Stokes parameter $I(r, f)$ of the MS HRS spectrum is shown in Fig. 5a. The peak of intensity (in red) centered

at 0 GHz between 9 and 10 km of altitude, corresponds to the position of the cloud. It is the contribution of the cloud particles (named by abuse of language "Mie contribution"). This spectrum is also characterized by the molecular contribution (Rayleigh contribution). The intensity of the spectrum below the cloud is lower than the intensity of the spectrum above the cloud due to particulate extinction. In Fig. 5b, the MS (computed with McRALI-FR) and SS (computed from SS theory) vertical profiles of spectral width are represented. We note a very good agreement between the SS theoretical and MS simulated values, both

at the cloudy and molecular levels. This suggests that MS effects have very little impact on spectral width. Then, in a second step, a software models a FP interferometer that separates the particulate contribution from the molecular contribution and provides the vertical profiles of particulate and molecular ATB as shown on Fig. 5c. The total ATB calculated directly from the spectrum, SS molecular and SS particulate backscatter profiles are also represented. Above the cloud, particulate ATB is not strictly zero and molecular ATB is not strictly equal to total ATB because of the FP remaining cross-talk effects (see

above). In the cloudy part between 9 and 10 km altitude, the molecular and particulate ATB logically decrease exponentially with the depth. The SS backscatter profiles decrease faster with depth than the MS backscatter profiles, revealing that MS

effects on ATB are not negligible. Under the cloud, molecular ATB is almost equal to total ATB. Particulate ATB is almost

zero. Some non-zeros values exist due to FP cross-talk effects but also due to Monte Carlo noise and MS processes.

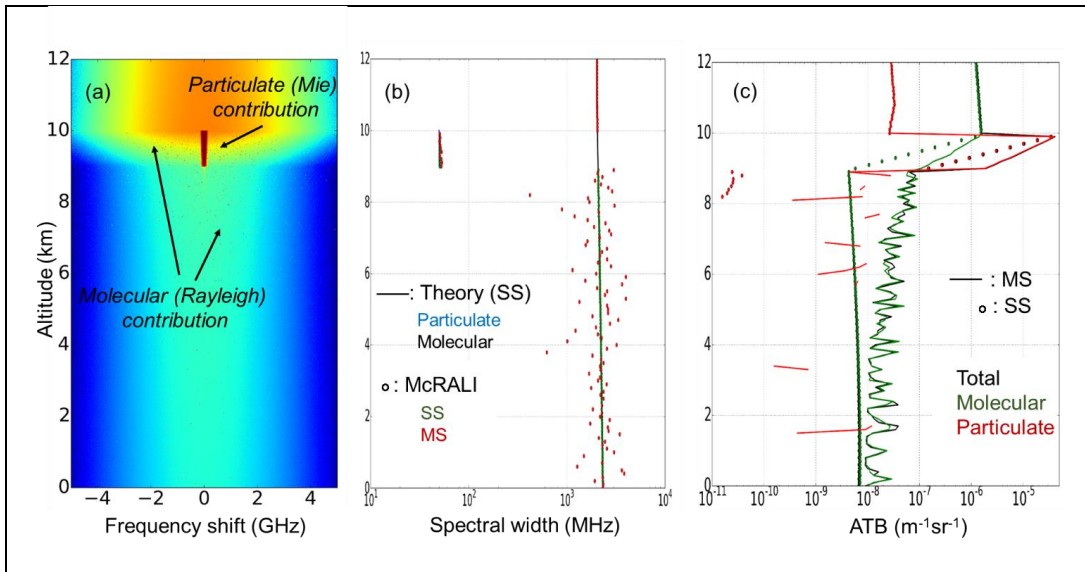

Figure 5: (a) Vertical profile of MS HSR spectrum for ATLID-like lidar. (b) Spectral width profiles. SS and MS spectral

width profiles computed by McRALI (circle) are in green and red, respectively. Theoretical SS molecular and SS particulate

width profiles (full line) are in black and blue, respectively. (c) Vertical profiles of MS (line) and SS (circle) backscattered

coefficient (ATB). Total, molecular and particulate signals are in black, green and red, respectively. The altitude of base of

the iced homogeneous cloud layer (optical depth of 3) is 9 km. Its geometrical depth is 1 km.

## 3. Assessment of errors induced by NUBF on lidar and radar data

The objectives of this section are to investigate effects of cloudy atmosphere having 3D spatial heterogeneities under multiple

scattering regime on HRS lidar and Doppler data by using McRALI-FR simulators. One of the simplest shapes of

heterogeneous cloud to study this kind of effects is the idealized "step" cloud defined in the international Intercomparison of

3D Radiation Codes (I3RC) phase 1 (Cahalan et al., 2005). The main interest is to model behaviour in the vicinity of the single

isolated jump in optical depth. With this in mind, we prefer to use an even more simplistic cloud model, the box-cloud,

described in the following paragraph. A detailed statistical analysis at different averaging scales of representative fine-structure

3D cloud field effects on lidar and radar observables is beyond the scope of this paper and will be investigated in a future work.

### 3.1. Conditions of simulation and definition of the box-cloud

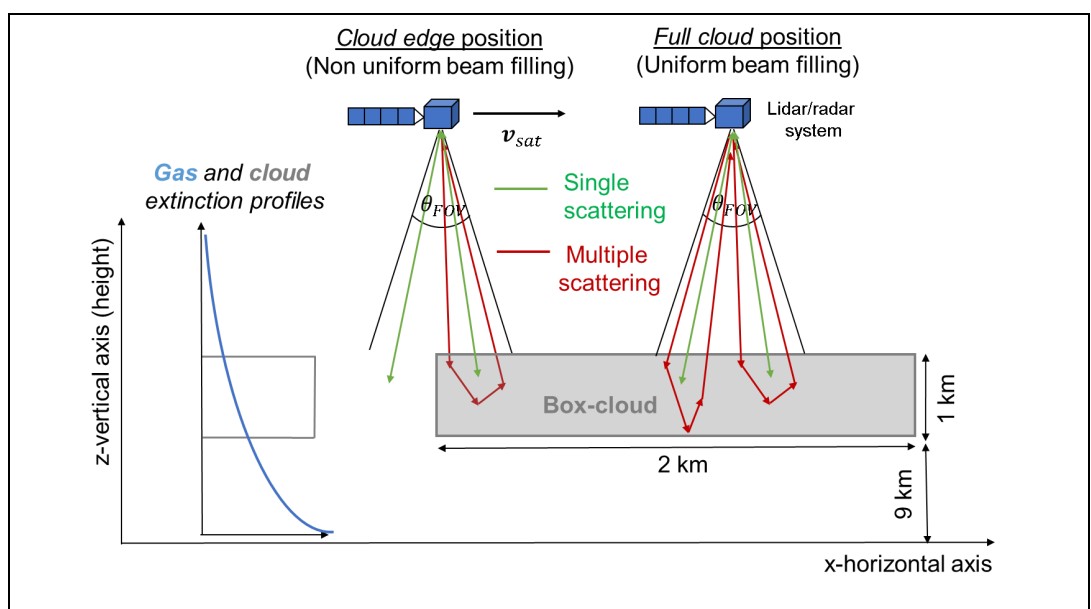

Figure 6: Schematic representation of two specific positions of spaceborne lidar/radar system relative to the idealized box-cloud. The box-cloud base altitude is 9 km, its geometrical thickness is 1 km, and its x-horizontal and y-horizontal extension is 2 km and infinite, respectively. Cloud vertical extinction profile is constant. In the two positions, single and multiple scattering photon paths examples are represented in green and red arrows, respectively.

The box-cloud base altitude is 9 km, its geometrical thickness is 1 km, and its x-horizontal and y-horizontal extension is 2 km and infinite, respectively. Temperature and pressure vertical profiles assume 1976 U.S. standard atmosphere models. Optical

5    cloud properties are characterized by the extinction coefficient set to 0.1, 1.0 and 3 km$^{-1}$.

Figure 6 shows a representation of two specific positions of spaceborne lidar/radar system relative to the idealized box-cloud. Cloud optical properties are spatially homogeneous within the box-cloud. When the lidar/radar system is just above the cloud edge, the NUBF effect can be significant whereas it is null when the system is completely over the cloud. Table 1 summarizes the conditions of McRALI-FR simulations of data from the HSR ATLID lidar and Doppler CPR radar of the EarthCARE

10   mission when the heterogeneous box-cloud is considered.

At a wavelength of 355 nm (lidar configuration), gas scattering properties are based on Hansen and Travis (1974). Gas Doppler broadening is computed assuming Maxwell-Boltzmann distribution as presented in Sect. 2.4.1. The scattering matrix was computed for a gamma size-distribution of ice crystals having an effective diameter of 50 μm and the aspect ratio of 0.2. The refractive index value was $= 1.3243 + i \cdot 3.6595 \cdot 10^{-9}$ ; the surface of particles was supposed to be rough (Yang and Liou,

1996). Optical characteristics were computed using the improved geometric optics method (IGOM) (Yang and Liou, 1996). The asymmetry parameter is $g = 0.73$, which is in agreement with experimental data for cirrus clouds (Gayet, 2004; Shcherbakov et al., 2006). Single scattering albedo is set to 1.0.

At 94 Ghz (radar configuration), we assumed Henyey-Greenstein phase function with asymmetry parameter $g = 0.6$. Single

scattering albedo is set to 0.98. These last two values are taken from Battaglia and Tanelli (2011) for the scenario involving a deep convective core with graupel. Wind vertical velocity (downdraft) is set to 6 ms$^{-1}$. We assume no wind turbulence nor particle sedimentation velocity. For a cloud layer at an altitude of around 9 km, the pressure, the temperature and the relative humidity can be set to 308 hPa, 229.7 K and 100%, respectively (1976 U.S. standard atmosphere); then gas absorption $\sigma_{abs} \approx 2 \times 10^{-5}$ km$^{-1}$. We assumed that this value is small enough to neglect the gas absorption for the simulations carried out in this

work.

Spacecraft velocity and altitude are set to $v_{sat} = 7.2$ km.s$^{-1}$ and 393 km, respectively. The lidar/radar system pointing angle is set to 0°. The lidar transmitter is assumed to be a Gaussian laser beam with $1/e$ angular half-width $\theta = 22.5$ µrad. The lidar receiver is assumed to be a top-hat telescope with a half-angle field of view $\theta_{FOV} = 32.5$ µrad, which represents a ground beam footprint of around 30 m. Radar transmitters and receivers are assumed to be Gaussian antenna with a 3-dB half-width

$\theta = \theta_{FOV} = 0.0475°$, which represents a ground beam footprint of around 660 m.

McRALI-FR code simulates the multiple scattering and single scattering idealized HSR and Doppler spectrum for lidar and radar configurations, respectively. Lidar spectra are computed for five positions (x-horizontal ground projected distance) relative to the box-cloud edge. Lidar positions values are $x = -8.6, -4.0, 0, 4.0$ and 8.6 m. Indeed, the ratio (we also talk about cloud coverage) of the cloudy part inside the ATLID lidar FOV divided by the full lidar footprint area at the altitude of 10 km,

is 10, 30, 50, 70 and 90 %, respectively. Then, a software computes apparent molecular and particulate backscattering coefficient profiles, assuming that ATLID/EarthCARE lidar is equipped with FP interferometers (see Sect. 2.5.3). For radar configuration, simulations are carried out every 100 m; Doppler spectra are computed for position values fixed at $x = -500, -250, 0, 250$ and 500 m. Then, a software computes reflectivity, Doppler velocity and Doppler velocity spectrum width profiles with the help of Eq. (16), (17) and (18).





|  | ATLID/EarthCARE type lidar | CPR/EarthCARE type radar |
|---|---|---|
| *Characteristics of lidar and radar systems* |  |  |
| Spacecraft altitude | 393 km | 393 km |
| Projected spacecraft velocity | 7.2 kms⁻¹ | 7.2 kms⁻¹ |
| Wavelength or frequency | 355 nm | 94 GHz |
| Pointing angle | 0° | 0° |
| Emitter model and beam half-width | Gaussian (1/e), 22.5 µrad | Gaussian (3-dB), 0.0475° |
| Receiver model and FOV half-angle | Top hat, 32.5 µrad (1) | Gaussian (3-dB), 0.0475° |
| Beam footprint | ~26 m | ~650 m |
| *Characteristics of cloudy atmosphere* |  |  |
| Temperature and pressure vertical profiles | U.S. standard atmosphere model (1976) | No gas (4) |
| Gas optical properties vertical profile | Hansen and Travis (1974) | No gas (4) |
| Gas Doppler broadening | Maxwell-Boltzmann distribution | - |
| Geometry of box-cloud model | x-wide = 2 km, y-depth = 100 km, z-thickness = 1 km | |
| Cloud top and base altitude | 9-10 km | |
| Cloud geometrical depth | 1 km | |
| Cloud extinction | 0.1, 1.0, 3 km⁻¹ | |
| Single scattering albedo | 1.0 | 0.98 |
| Cloud phase function | Rough ice crystals, $R_{eff} = 25$ µm (Yang and Liou, 1996) | Henyey-Greenstein |
| Asymmetry parameter | 0.73 | 0.6 |
| Interferometer | Fabry-Pérot | - |
| Vertical wind velocity | 0 ms⁻¹ | 6 ms⁻¹ (downdraft) (5) |
| Wind turbulence (standard deviation $\sigma_{turb}$ of Gaussian isotropic model) | 0 ms⁻¹ | 0 ms⁻¹ |
| Particle sedimentation velocity | 0 ms⁻¹ | 0 ms⁻¹ |
| *Simulated quantities from idealized range and frequency resolved Stokes parameters (2)* |  |  |
| Relative horizontal position of lidar/radar system to the cloud edge | x = -8.6, -4.0, 0, 4.6 and 8.6 m | x = -500, -250, 0, 250 and 500 m (7) |
| Power spectrum profiles | High spectral resolution spectrum | Doppler spectrum |
| Vertical profiles | Molecular, particle and total backscatter, depolarization ratio | Doppler velocity, Doppler spectral width, reflectivity |
| Vertical resolution (3) | 100 m | 100 m |
| Power spectrum interval resolution | 0.01 Hz | 1 ms⁻¹ |

Table 1: Description of the simulation conditions presented in this work. This table summarizes the characteristics of the

ATLID/EarthCARE type lidar and the CPR/EarthCARE type radar and properties of cloudy atmosphere and quantities




computed by McRALI-FR codes. (1) Other simulations are performed with FOV half-angle of 325 µrad. (2) Idealized means that receiver noise, along-track integration and Nyquist folding are ignored. (3) ATLID and CPR vertical resolution is 100 m from -1 to 20 km in height. (4) Gas absorption (Liebe, 1985) can be taken into account. (5) A specific case with a two-layer cloud with 6 m/s and – 6m/s vertical wind velocity in the top layer and bottom layer, respectively, is also studied. (7) For radar configurations, simulations are also carried out every 100 m according to the horizontal distance.



### 3.2. CPR/EarthCARE configuration

### 3.2.1. NUBF effects on Doppler radar data: Doppler spectrum and reflectivities

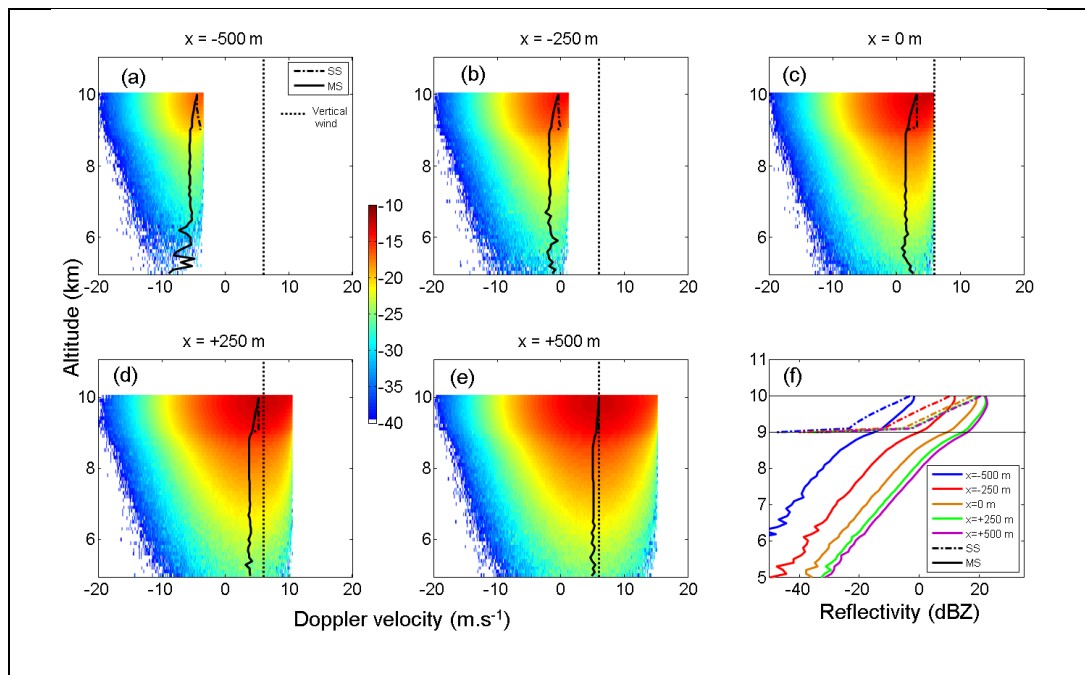

Figure 7: Vertical profiles of MS radar Doppler spectra (logarithm of the spectral density (in m$^{-1}$.sr$^{-1}$.(m.s$^{-1}$)$^{-1}$)) in CPR/EarthCARE configuration corresponding to the five positions $x = -500$ m (a), $x = -250$ m (b), $x = 0$ m (c), $x = +250$ m (d) and $x = +500$ m (e) of the satellite relative to the edge of the box-cloud. SS (black line) and MS (black dotted line) vertical profiles of Doppler velocity are superimposed. Vertical wind velocity profile (downdraft) fixed at $w_{wind} = 6$ m.s$^{-1}$ during our simulation is also drawn (black dotted line). (e) The five reflectivity (in dBZ) profiles (MS: full lines, SS: dotted lines) corresponding to the five positions ($x = -500$ in blue, $x = -250$ in red, $x = 0$ m in brown, $x = 250$ m in green and $x = 500$ m in magenta) relative to the edge of the box-cloud. Cloud optical depth is 3.

5  Figures 7a, b, c, d and e show vertical profiles of MS radar Doppler spectra density in CPR/EarthCARE configuration corresponding to the five positions of satellite relative to the edge of the box-cloud with optical depth set to 3. Regardless of the satellite position, Doppler spectra correspond to negative Doppler velocity values. This is consistent with the convention that the Doppler velocity is positive for motion away from the radar. As the satellite approaches the edge of the cloud and carries on, the NUBF effect decreases. Indeed, the Doppler spectrum becomes more and more symmetrical. The asymmetric

10  shape of the Doppler spectrum is due to zero values beyond a critical value of Doppler velocity $v_{crit}$. As an example, $v_{crit} \approx$





$-3.4$ ms$^{-1}$ in Fig. 7a. The explanation of $v_{crit}$ value is purely geometric. The Doppler broadening is dominated by the Doppler fading due to satellite motion. Under SS approximation, neglecting wind velocity, Doppler shift is given by $\Delta v = \frac{1}{2} \boldsymbol{v}_{sat}.(\boldsymbol{k}_0 - \boldsymbol{k}'_1)$, with $\boldsymbol{k}_0 = -\boldsymbol{k}'_1$. Assuming $\boldsymbol{v}_{sat}$ with x-horizontal positive component, $\boldsymbol{k}_0$ in the (z-x) vertical plan with $k_x$ x-horizontal component, then $\Delta v = -v_{sat}.k_x$. Assuming that satellite is at the x-horizontal $d$ distance to the box-cloud

edge and at the z-vertical $D$ distance above the cloud, with a vertical (downdraft) wind velocity $v_{wind}$ fixed at 6 ms$^{-1}$, then

$v_{crit} = -v_{sat}\, d/(d^2 + D^2)^{1/2} + v_{wind}$. For $x = -500$, $x = -250$, $x = 0$, $x = 250$ and $x = 500$ m, $v_{crit} = -3.4$, $v_{crit} = -1.3$, $v_{crit} = 6$, $v_{crit} = 10.7$ and $v_{crit} = 15.4$ m.s$^{-1}$, respectively. These values are very close to those estimated from the five respective power spectra of Fig. 7.

Figure 7f also shows the five reflectivity profiles corresponding to the five positions of the satellite relative to the edge of the

box-cloud. MS reflectivity profiles are larger than SS reflectivity profiles because MS processes logically increase the reflectivity value. We can also see MS effects on the apparent reflectivity that is non-null under the cloud layer, contrary to the SS apparent reflectivity. At the same time, as the satellite approaches the edge of the cloud and keeps moving forward, SS and MS apparent reflectivity profiles values increase, due to the fact that the NUBF effect decreases. Many studies have focused on the NUBF effect on rain fields retrieved by radar from space (Amayenc et al., (1993), Testud et al., (1996); Durden

et al., (1998); Iguchi et al., (2009)). Iguchi et al., (2000) showed that the NUBF effect could be accounted for by a factor determined from horizontal variation of attenuation coefficient. Our simulations are coherent with literature. Detailed investigation of the NUBF effect on reflectivity profiles for spaceborne cloud radar will be carried out in a later work.

### 3.2.2. NUBF effects on Doppler velocity and Doppler spectrum width

Figure 7 shows the SS and MS vertical profiles of Doppler velocity superimposed on the five power spectra for the five

positions of the satellite relative to the box-cloud edge $x = -500,\ -250,\ 0,\ 250$ and $500$ m. Since each power spectrum has no value beyond the $v_{crit}$ velocity, which is due to the NUBF effect, it is obvious that the profile of the apparent Doppler velocity is different from the profile of the vertical wind velocity fixed at 6 ms$^{-1}$. Figure 8 shows the MS and SS apparent Doppler velocity and Doppler spectrum width computed every 100 m along the horizontal axis. These quantities are estimated at different altitudes (cloud top, middle and base) and are plotted as a function of the satellite distance to the box-cloud left

edge. Differences between apparent Doppler velocities are in general small (around 1 m.s$^{-1}$) whatever the altitude is, and differences between MS and SS Doppler velocities are also small, no larger than 1 m.s$^{-1}$ at the bottom of the cloud. The same conclusions can be drawn for Doppler spectrum width where differences are no larger than 0.3 m.s$^{-1}$. This implies that MS processes do not play an important role in the estimation of the apparent Doppler velocity nor in the estimation of apparent spectrum width (for the specific conditions of simulation with the box-cloud) compared to the NUBF effect. The NUBF

Doppler velocity bias between apparent Doppler velocity and "true" vertical wind velocity fixed at 6 m.s$^{-1}$ is around -10, -5, -3, -2 and -1 ms$^{-1}$ at $x = -500, -250, 0, 250, 500$ m, respectively.

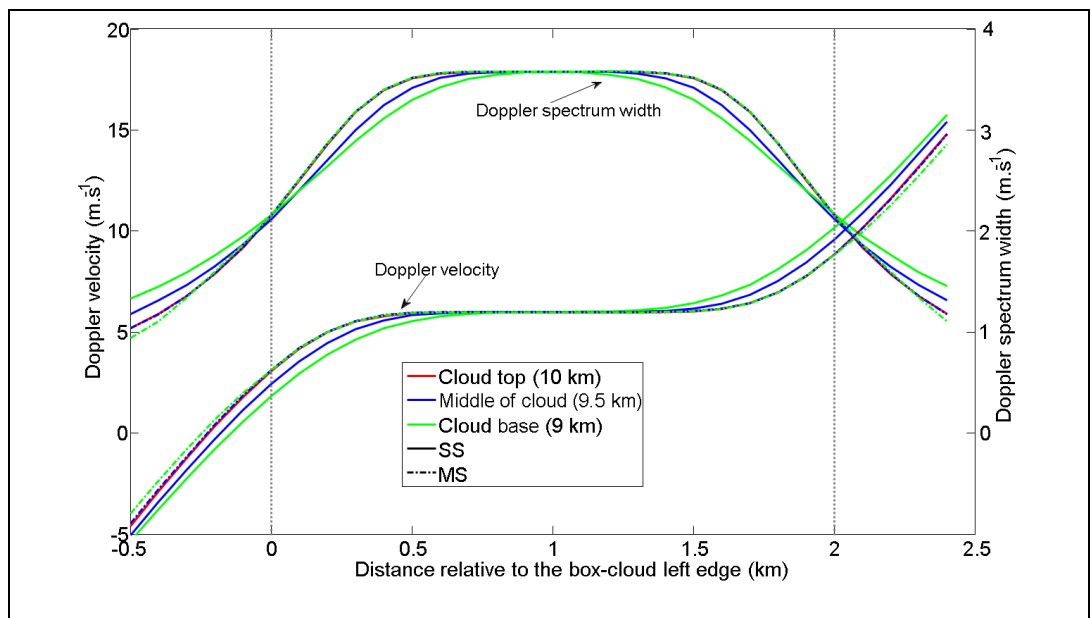

Figure 8: MS (full lines) and SS (dotted line) apparent Doppler velocity and Doppler spectrum width as a function of the distance of the satellite relative to the box-cloud left edge. Values are computed at cloud top (10 km of altitude, in red), middle (9.5 km of altitude, in blue) and base (9 km of altitude, in green). Optical thickness of the box-cloud is 3. Simulations are done every 100 m.

In general NUBF bias of Doppler velocity can be expressed as a function of the distribution of the radar reflectivity (Tanelli et al., 2002). An estimate of NUBF bias of Doppler velocity can be obtained by considering the difference between the Doppler velocity computed with a satellite velocity (i.e. $v_{sat} = 7.2$ kms$^{-1}$) and the Doppler velocity computed with a satellite velocity

5  set to 0 m.s$^{-1}$ (Battaglia et al., 2018). Sy et al. (2014) showed that the NUBF bias of Doppler velocity is correlated to the horizontal gradient of reflectivity and demonstrated that theoretical proportional coefficient $\alpha$ value is bounded between 0.165 and 0.219 m.s$^{-1}$(dBZ.km$^{-1}$)$^{-1}$. Kollias et al. (2014) estimated this proportional coefficient value close to $\alpha = 0.23$ m.s$^{-1}$(dBZ.km$^{-1}$)$^{-1}$ for along-track horizontal integration of 500 m (i.e. Doppler CPR/EarthCARE resolution) and for all their available simulations performed with a cirrus cloud and a precipitation system. Figure 9 shows Doppler velocity NUBF bias

10  as a function of horizontal reflectivity gradient for horizontal integration of 500 m, for four positions relative to the box-cloud edge (-200 m, 100 m, 0 m and 100 m). Computations are carried out for different optical depths of the box-cloud are 0.1, 1 and 3. We note that the $\alpha$ value is between 0.14 and 0.16 m.s$^{-1}$(dBZ.km$^{-1}$)$^{-1}$ and is almost independent of the position of satellite relative to the box-cloud edge. If satellite position is just above the cloud edge, the proportional coefficient value is close to 0.15 m.s$^{-1}$(dBZ.km$^{-1}$)$^{-1}$, a value close to value obtained by Sy et al. (2014).

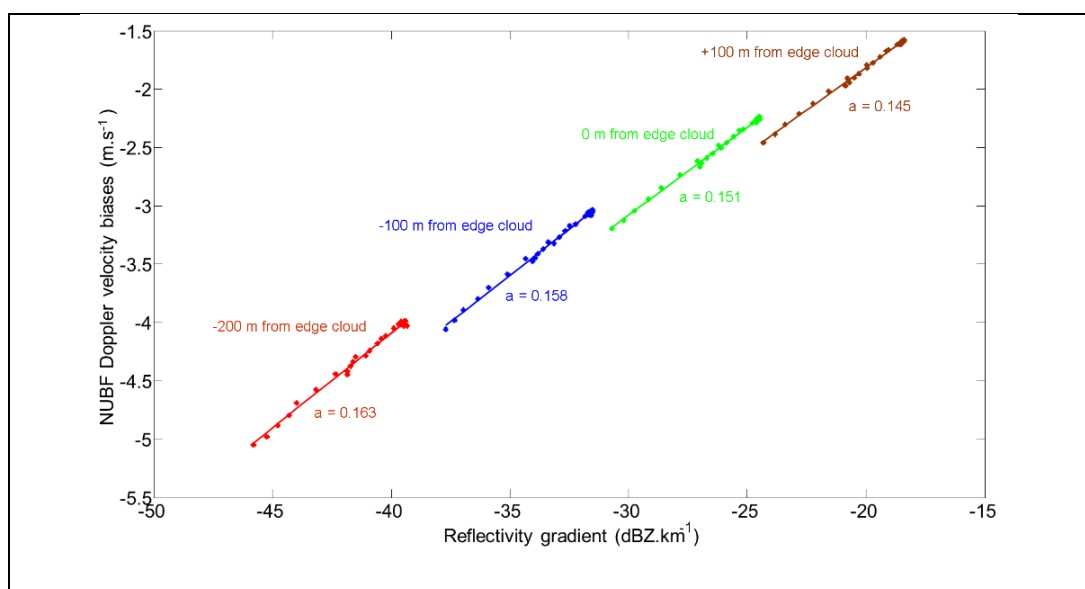

Figure 9: NUBF velocity bias as a function of horizontal reflectivity gradient for horizontal integration of 500 m, estimated for four positions relative to the box-cloud edge: -200 m (red), -100 m (blue), 0 m(green) and +100m (brown). Optical depths of box-cloud are 0.1, 1 and 3. Vertical (downdraft) velocity is set to 6 m.s$^{-1}$. Proportional coefficient value $\alpha$ (in m.s$^{-1}$(dBZ.km$^{-1}$)$^{-1}$) between NUBF velocity bias and horizontal reflectivity gradient is also given.

### 3.2.3. Effects of vertically heterogeneous wind velocity on Doppler velocity

5    A first study of the effects of multiple scattering on the Doppler velocity vertical profile in case of vertically heterogeneous wind velocity is carried out, for a very specific case, in CPR/EarthCARE configuration. Indeed, for a homogeneous cloud layer with a base altitude of 9 km and with geometrical thickness of 1 km, the vertical velocity is set to 6 m.s$^{-1}$ (downdraft) and to -6 m.s$^{-1}$ (updraft) in the upper and the lower part of the cloud layer, respectively. Figure 10 shows vertical profiles of MS radar Doppler spectrum and the MS and SS Doppler velocity profiles computed with McRALI-FR. The measured Doppler velocity

10   under SS regime (black dotted line in Fig. 10) is equal to 6 m.s$^{-1}$ (-6 m.s$^{-1}$) in the upper (lower) part of the cloud, the SS Doppler velocity can be used as the reference of the true velocity. In the upper part of the cloud, measured MS Doppler velocity is 6 m.s$^{-1}$, equals the true velocity. In the lower part of the cloud, the measured MS Doppler velocity is biased by multiple scattering processes with a value not smaller than -3 m.s$^{-1}$, contrary to the true velocity of -6 m.s$^{-1}$ in this cloudy part. For altitudes less than 7 km, we can also note that multiple scattering processes can lead to a Doppler velocity lower than - 6 m.s$^{-1}$.



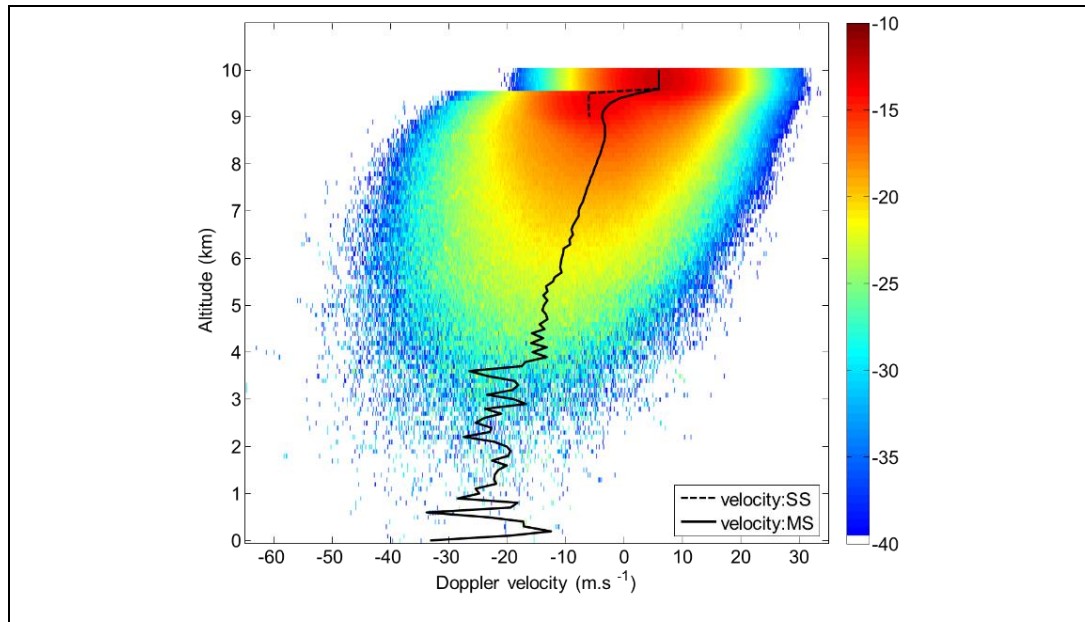

Figure 10: Vertical profiles of MS radar Doppler spectrum (logarithm of the spectra density (in $m^{-1}.sr^{-1}.(m.s^{-1})^{-1}$)) in CPR/EarthCARE configuration simulated by McRALI-FR. The MS Doppler velocity (black line) and SS (black dotted line) vertical profiles of Doppler velocity are superimposed. The homogeneous cloud layer base altitude is 9 km, its geometrical thickness is 1 km. The vertical velocity is set to 6 $m.s^{-1}$ (downdraft) and to -6 $m.s^{-1}$ (updraft) in the upper and the lower part of the cloud layer, respectively. Optical depth is 3.

### 3.3. ATLID/EarthCARE configuration

#### 3.3.1. NUBF effects on the HRS lidar data

In order to investigate the NUBF effects on HSR lidar observables under MS regimes we firstly compare simulation results

5   carried out with the box-cloud (full 3D simulation) of optical depth equal to 3 (hereafter called 3D cloud) with simulations performed under the Plane-Parallel and homogeneous cloud model (hereafter called PP cloud) and under the Independent Column Approximation (or independent pixel approximation) cloud model (hereafter called ICA cloud). PP theory and ICA assumption are commonly used to assess the radiative effects of inhomogeneous cloud when cloud unresolved variability and net horizontal fluxes are ignored, respectively (Marshak and Davis, 2005). For the specific case of the satellite position relative

10   to the box-cloud edge is $x = -4.0$ m (see Fig. 11a), the cloud coverage $\alpha$ inside the lidar receiver FOV is 30% (see Sect. 3.1). As cloud optical depth (COD) is set to 3, this implies that mean COD weighted by cloud coverage inside the lidar footprint is 0.9, the assigned value to the optical depth of the PP cloud (Fig. 11b). In other words, PP profile can be considered as a profile computed with a homogeneous cloud with optical depth equal to the mean optical depth of the cloudy part weighted by the





cloud cover of the 3D cloud. The ICA simulation is carried out by averaging 30 % of a simulation with a homogeneous cloud with COD of 3 and 70% of a simulation in a clear sky atmosphere (Fig. 11c). In other words, ICA profiles can be considered as a profile averaged over columns (two columns in this case) weighted by the cloud coverage.

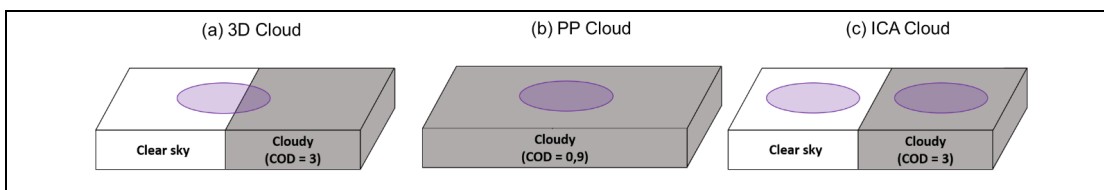

Figure 11: Conceptual representation of cloud models used for the HSR lidar simulations: (a) box-cloud model (3D), (b) Plane-Parallel and homogeneous (PP) model and (c) Independent Column Approximation (ICA) cloud. The purple circles represent the lidar footprint location for the simulations.

Figure 12 shows the HRS spectra simulated by McRALI-FR for the three cloud models at four altitude levels: above the cloud (12 km, Fig. 12a), at cloud top (9.8 km, Fig. 12b) and base (9.2 km, Fig. 12c) and below the cloud (8.5 km, Fig. 12d). In the clear sky region above the cloud (Fig. 12a), the three frequency spectra line up very well, as expected, because the lidar laser beam has not yet been scattered by the cloud. A similar feature is observed at cloud top (Fig. 12b), whereas a few spikes can

10  be seen on the molecular broad spectrum. Deeper in the cloud (Fig 12c) and below (Fig. 12d), spikes are more numerous with higher intensity. These spikes are simulation artefacts. They are caused by specific events of multiple scattering, namely, by the cases when forward scattering is involved during a photon random path. For example, the photons can be first scattered by air molecules inducing a large frequency shift, then by cloud particles inducing a large contribution due to the highly forward-peaked phase function. The scattering phase function of ice particles spans about six orders of magnitude. Thus, the forward

15  scattered photons have a weight, which is several orders of magnitude larger than those scattered in other directions. At the same time, such cases occur rarely. Consequently, we should carry out simulations with an unrealistic number of photons emitted by the lidar to smooth spikes, which is not possible. Spikes are not observed on simulations under the single scattering regime (not shown here).



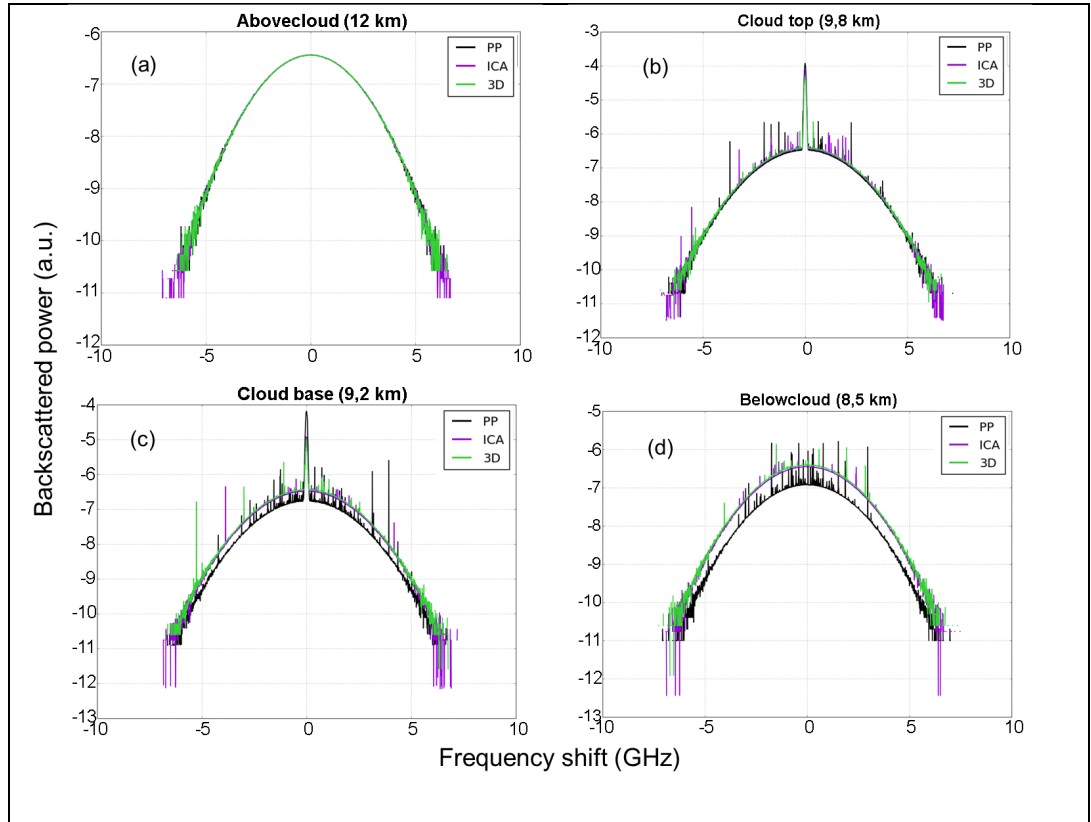

Figure 12: Normalized HSR power spectra (a) above the cloud (12 km of altitude), (b) at cloud top (9.8 km of altitude), (c) at cloud base (9.2 km of altitude), and (d) below the cloud (at 8.5 km of altitude). Black lines represent simulations using the PP cloud model, purple lines the ICA cloud model and the green lines the 3D box-cloud model.

One can clearly see that the intensity of the central Mie (particulate) peak computed by ICA and 3D is lower than the one computed by the PP simulation. The opposite behaviour is observed concerning the Rayleigh (molecular) scattering region: the broad and low intensity spectra for ICA and 3D simulations show larger values (in intensity) compared to PP simulation, and for the full range of frequency shift. The same observation can be made below the cloud.

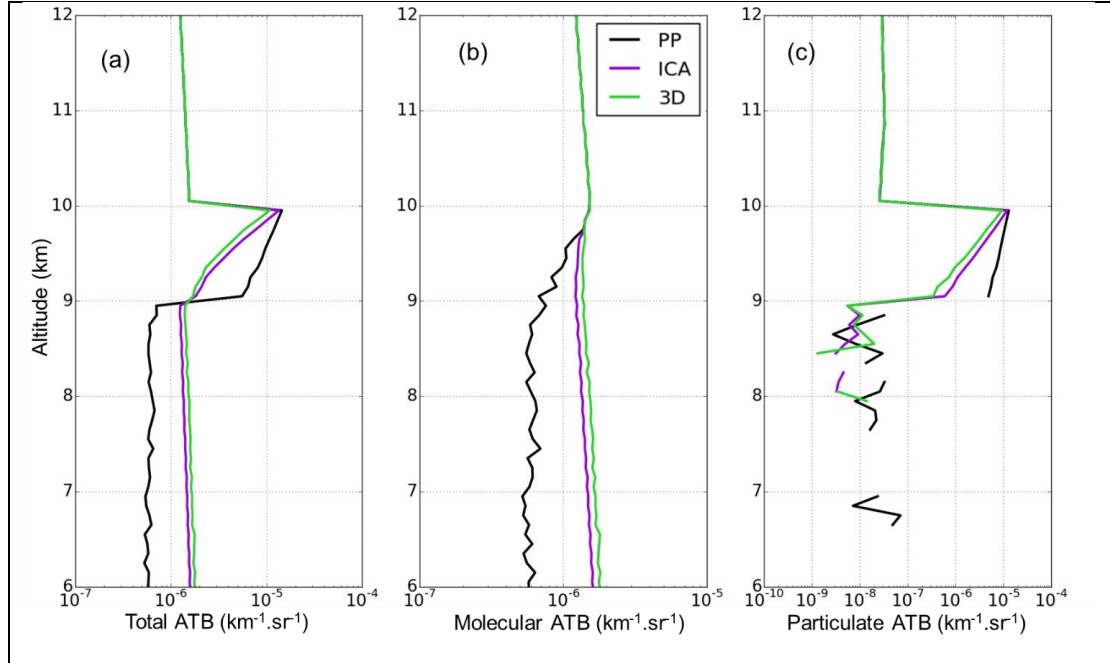

Figure 13: Vertical profiles of (a) total, (b) molecular and (c) particulate ATB simulated by McRALI. Cloud is located between 9 and 10 km. Black lines represent simulations using the PP cloud model, purple lines the ICA cloud model and the green lines the 3D box-cloud model.

In order to obtain the molecular and particulate ATB vertical profiles for the three cloud models (Fig. 13b and Fig. 13c), the HSR spectra are filtered by a modelled Fabry-Perot interferometer (see Sect. 2.5.3 for the filtering parameters) at each altitude level. The ATB vertical profiles (Fig. 13a) exhibit features observed on HSR spectra more clearly: once the cloud is reached (i.e. below 10 km), the three cloud models give very different total ATB. Higher values are observed in clouds for the PP model compared to ICA and 3D cloud models, whereas the opposite is observed below the cloud. This feature shows that PP cloud representation can lead to large discrepancies and it is suitable to account for 3D cloud structure. On the contrary, ICA cloud models give results rather close to 3D cloud models. In Fig. 13b and Fig. 13c the PP cloud shows the most significant differences, with PP particulate ATB in cloud larger than that from ICA and 3D particulate ATB and a PP molecular ATB smaller. To a lesser extent, ICA and 3D computations show differences also with 3D total and 3D particulate ATB smaller than ICA computation. This difference is the opposite for the molecular ATB.

In order to quantify the differences coming from the cloud models, PP, ICA and 3D biases have been computed. These biases, well described in Davis and Polonsky (2005), were firstly defined in the radiance framework by Cahalan et al. (1994) and were





adapted to lidar signals framework by Alkasem et al. (2017). The 3D bias on ATB (i.e. $\Delta ATB_{3D}$) is the sum of the PP bias (i.e. $\Delta ATB_{PP}$) and of the ICA bias (i.e. $\Delta ATB_{ICA}$) defined as:

$$\Delta ATB_{PP} = ATB_{PP} - ATB_{ICA} \tag{22.1}$$

$$\Delta ATB_{ICA} = ATB_{ICA} - ATB_{3D} \tag{22.2}$$

$$\Delta ATB_{3D} = \Delta ATB_{PP} + \Delta ATB_{ICA} = ATB_{PP} - ATB_{3D} \tag{22.3}$$

where $ATB_{ICA}$, $ATB_{PP}$ and $ATB_{3D}$ are ATB computed by McRALI with the ICA, PP and 3D cloud models, respectively.The relative biases are also computed and correspond to the biases divided by the reference $ATB_{3D}$. In appendix B, it is shown that

5   the PP bias of molecular ATB and particulate ATB is always negative and positive, respectively. It is also shown that the larger multiple scattering is the smaller the PP particulate bias. Note that the PP bias of total ATB is positive but becomes negative with increasing cloud optical depth (Alkasem et al., 2017).

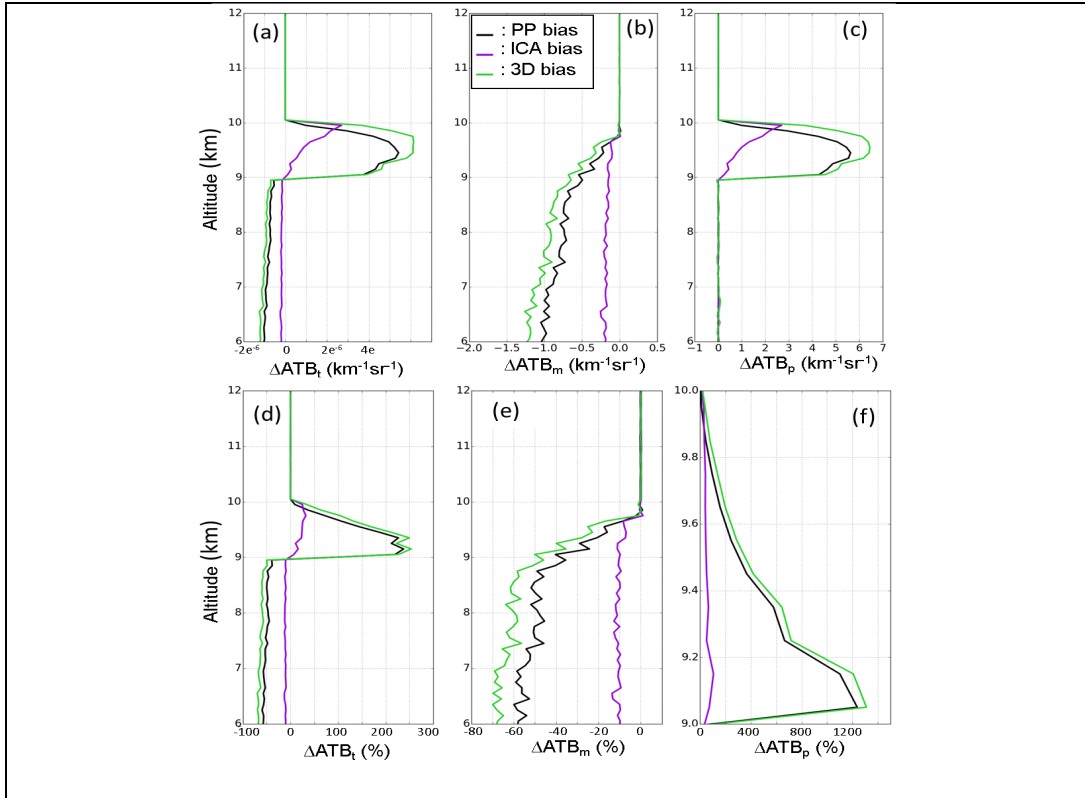

Figure 14: Vertical profiles of biases on (a) total, (b)molecular and (c) particulate ATB and vertical profiles of relative biases on (d) total, (e) molecular and (f) particulate ATB. Black lines represent simulations using the PP cloud model, purple lines the ICA cloud model and the green lines the 3D box-cloud model.





Figure 14 shows that the PP biases are the largest both on total ATB and the molecular and particulate components. Indeed, it reaches 250%, -60% and 1200% for total, molecular and particulate ATB respectively. The ICA biases present lower values (around 25%, -10% and less than 100% for total, molecular and particulate ATB respectively). These results show thus that

5 3D biases are mainly due to the PP biases.

Based on the three cloud models, further simulations have been carried out with varying the cloud coverage inside the lidar FOV from 10 % to 90% in order to evaluate the impact of cloud coverage on HSR lidar observations. It has been carried out for the study of the NUBF effect on radar observations in a similar way (see Sect. 3.2). The SS bias and the MS relative bias on total, molecular and particulate ATBs at cloud top (9.8 km), cloud base (9.2 km) and in the middle of the cloud (9.5 km)

10 have been computed for each cloud model in Fig. 15.

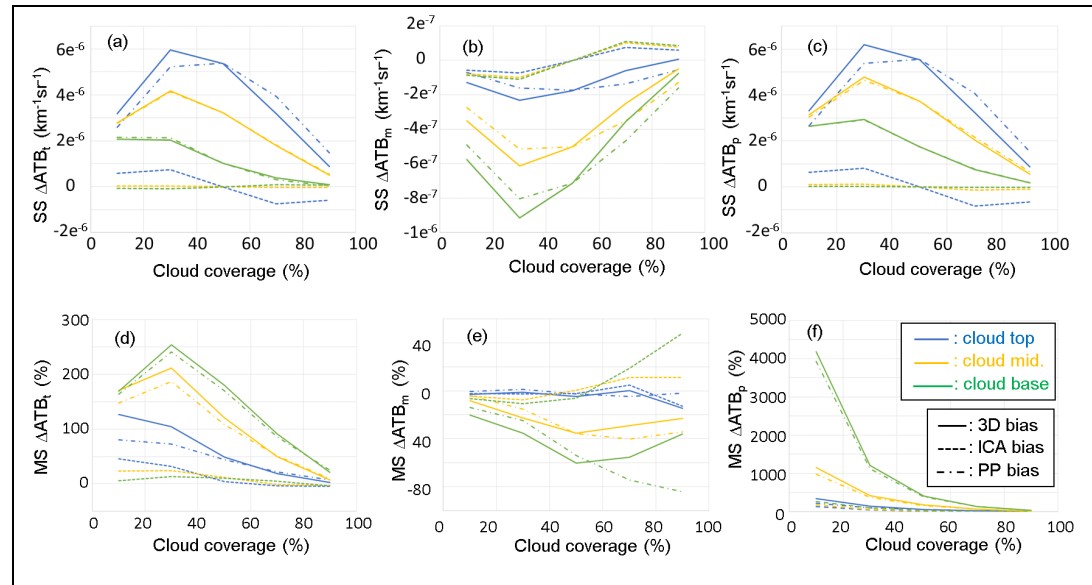

Figure 15: ICA (dashed), PP (dotted dashed) and 3D (full line) biases under single scattering (SS) regime on (a) total ATB, (b) molecular ATB and (c) particulate ATB as a function of cloud coverage (%) inside the lidar receiver FOV, computed at cloud top (9.8 km, blue curves), in the middle of the cloud (9.5 km; yellow curves) and at cloud base (9.2 km; green curves). (d), (e) and (f) same as (a), (b) and (c) but for relative bias and MS regime. Cloud optical depth is 3.

Figure 15d and Fig. 15f show that MS total and particulate biases decrease when cloud coverage increases and reach almost zero for 90% cloud coverage. These relative biases always show the largest values at cloud base, then in the middle of cloud,



then at cloud top whereas it is not observed for SS bias (Fig. 15a and Fig. 15c). This is due to the division by reference $ATB_{3D}$ values that exponentially increase with cloud altitude. We also find that PP bias are positive, with maximum bias of 250% observed for the total relative ATB at cloud base and for a cloud coverage of 30%. These last two observations are consistent with simulation results done under SS regime: PP biases of total ATB and of particulate ATB are generally positive (see Eq.

(B8)) and strictly positive (see Eq. (B7)), respectively. The maximum bias of ATB also occurs for a cloud coverage of 30%. Otherwise, regardless of cloud coverage, Fig. 15d and Fig. 15f show that MS total and particulate ICA biases are still rather small (< 50% and <200% respectively) compared to the PP biases (< 250% and <4000% respectively). For total and particulate ATB, PP bias are the largest and are mainly responsible for the 3D biases, confirming the findings of the previous section, and that for any cloud coverage, both in SS and MS regime.

For molecular signals, Fig. 15e shows the negative PP bias that increases in absolute value with increasing the cloud coverage, reaching -40% and - 80% for a 90% cloud coverage in the middle of the cloud and at cloud base respectively. The negative value of MS molecular PP bias is consistent with SS molecular PP bias definition (see Eq. (B6)). Otherwise, MS ICA bias is negative for a cloud coverage smaller than 50 % and rather positive for a cloud coverage larger than 50 %. This latter observation is consistent with simulation results performed under SS regimes (see Fig 15b): molecular ICA bias is negative,

null and positive for a cloud coverage smaller than, equal to, and larger than 50%, respectively. Figure 15e shows that MS ICA bias reaches only 15% and 50% in the middle of the cloud and at cloud base respectively. At cloud top, all the molecular biases remain small (between 5% and -15%). Because of the competition under MS regime between the rather negative PP bias with the positive and negative ICA bias of the same order of magnitude, no specific trend on the 3D molecular biases according to the cloud coverage can be highlighted and conclusions are less obvious than for total and particulate ATB.

Finally, our simulation results also show that 3D bias decreases in magnitude when COD decreases, due to the PP bias that decreases. For example, if COD is 0.1, 3D total, molecular and particulate biases are less than 6%, 15 % and 100%, respectively, mainly driven by ICA bias.

### 3.3.2. Impact of size of field of view on total, molecular and particulate ATB

We briefly investigate the impact of the size of the field of view (FOV) by carrying out simulations with a FOV ten times

greater than that one in the simulations performed in the previous sections. Simulations have been carried out for a cloud coverage of 50%, implying COD of 1.5 for the PP cloud model. Figure 16 shows vertical profiles of ATB biases and relative ATB biases computed with this large FOV (i.e. 650 µrad) and those computed with the ATLID FOV (i.e. 65 µrad).

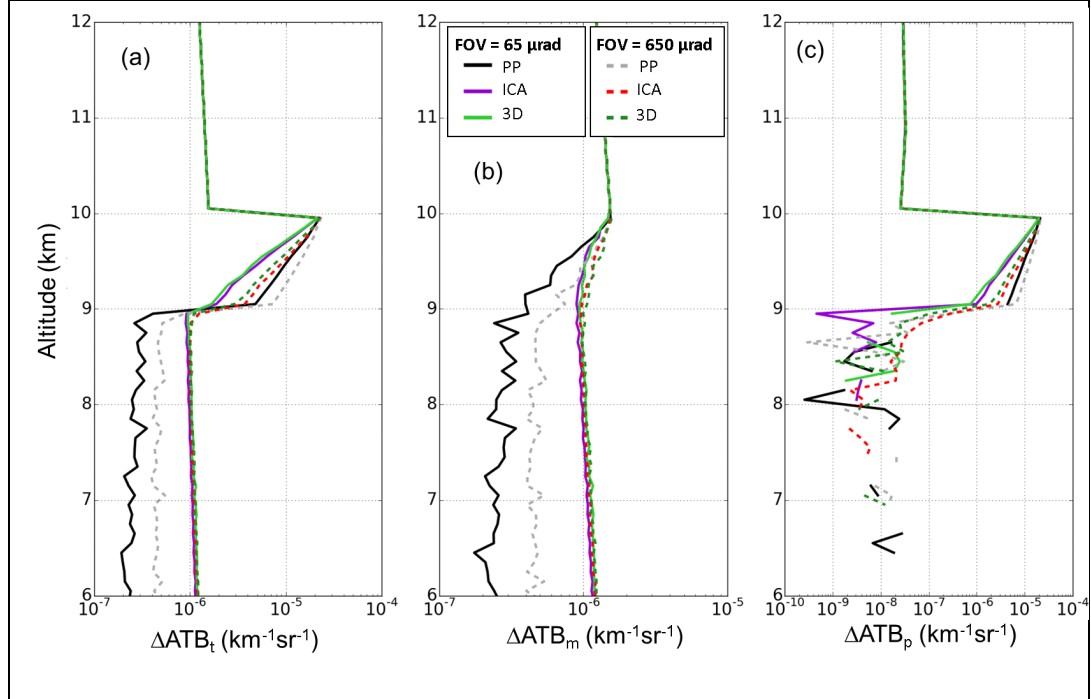

Figure 16: Same as Fig. 13 but with 50% cloud coverage. Simulations with ATLID FOV (65 µrad) are in full lines and simulations with a 650 µrad FOV are in dotted lines.

When comparing ATB computed with the three cloud models (i.e. PP, ICA and 3D cloud model) for the large FOV, the same conclusions as for the ATLID FOV can be made: when reaching the cloud base, total and particulate ATBs show lower values for ICA and 3D models than for PP model, and the opposite for the molecular ATB. Biases also show the same trend as for

5   ATLID FOV with a maximum bias for the PP model, and lower values for ICA models.

When comparing simulations carried out with a large FOV with those performed with a small FOV, we observed that with a larger FOV, ATB are larger when going into the cloud, and this for the three cloud models. The reason is mainly the multiple scattering which is obviously more pronounced as the FOV increases. Figure 16 shows that the biases and the relative biases for the large FOV present the same trend as for a small FOV. For total and particulate ATB, we can note that ICA biases are

10   larger whatever the vertical position in the cloud whereas PP biases are smaller in the upper part of the cloud, due to multiple scattering which becomes more significant as the FOV increases; this latter behaviour is coherent with Eq. (B7). The relative biases for the large FOV show slightly smaller values for the three signal components (total, molecular and particulate). The maximum values for the 3D bias are 150%, -60% and 200% for the total, molecular and particulate ATBs respectively.



Figure 17: Same as Fig. 14 but with a 50% cloud coverage. Simulations with ATLID FOV (65 µrad) are in straight lines and simulations with a 650 µrad FOV are in dotted lines.





### 4. Conclusions

This paper presents the Monte-Carlo code McRALI-FR that provides simulations of range $z$ and frequency $f$ resolved Stokes parameters $\mathbf{S}(z, f) = (I, Q, U, V)$ recorded by different kinds of monostatic polarized high spectral resolution lidars as well as Doppler radars from 3D cloudy atmosphere and/or precipitation fields. McRALI-FR is an extension of 3DMcPOLID, a Monte

Carlo code simulating polarized active sensor (Alkasem et al., 2017). The core of McRALI-FR is based on the 3D polarized Monte Carlo atmospheric radiative transfer model 3DMCPOL (Cornet et al., 2010; Fauchez et al., 2014) which uses the local estimate method to reduce the noise level. Gas absorption in micro-wavelength is taken into account according to the work of Liebe (1985). McRALI-FR considers the Doppler effect related to the motion of hydrometeors and aerosols. The random motion, i.e., the turbulent flow, is supposed to be homogeneous and isotropic. It is modeled as a multivariate normal

distribution. Generally, the regular motion of particles, i.e., the wind and/or precipitating hydrometeors, is assigned as a 3D vector field. The spectral distribution of the molecular scattering is modeled following the conventional method based on the Doppler shift from independent molecules moving with a Maxwell distribution of velocities. Each of the Stokes parameters is computed by McRALI-FR as a two-dimensional matrix (range and frequency resolved) and stored in an output file. Separated software uses the saved files to account for spectral and polarization characteristics of receivers and computes profiles of

corresponding HSR lidar or Doppler radar signals.

A study has been carried out on the effects of NUBF on the HSR ATLID lidar and Doppler CPR radar signals of the EarthCARE mission with the help of the academic 3D box-cloud, characterized by a single isolated jump in cloud optical depth. It is the simplest 3D cloud model that can be used to show and interpret the 3D radiative effects of clouds and for which the displayed results can only be obtained if the simulator is entirely in 3D. Moreover, for simplification, the wind speed is

assumed only vertical and constant. Particles sedimentation velocity is null.

Regarding Doppler CPR radar signals, it appears that multiple scattering does not affect the velocity estimation when the cloud characteristics are locally homogeneous across the radar beam. But if vertical wind velocity sharply varies with altitude, measured Doppler velocity profile can be largely affected by multiple scattering processes, as already mentioned by Battaglia and Tanelli (2011). At the same time, it is confirmed that the horizontally non-uniform beam filling induces a severe bias in

velocity estimates. Indeed, Doppler spectra shape is geometrically affected by the NUBF: the shape is all the more asymmetrical as the radar system vertically points away from the edge of the box cloud, inducing a bias in the estimation of the Doppler velocity. Within our very specific conditions of simulation with the box-cloud and with McRALI-Fr code, we found a proportional coefficient value around 0.15 m.s$^{-1}$(dBZ.km$^{-1}$)$^{-1}$ close to that obtained by Sy et al. (2014) and Kollias et al. (2014).

Regarding HSR ATLID lidar signals, we confirm that multiple scattering processes are not negligible, whatever the box-cloud cloud optical depth between 0.1 and 3, as previously studied by Reverdy et al. (2015) and pointed out by Donovan (2016). We

also investigated the NUBF effect due to different cloud coverages inside the FOV on HSR ATLID lidar observables under MS regime. For this purpose, we computed the vertical profiles of the 3D, PP and ICA biases for total, molecular and particulate ATB. The main conclusion is that 3D biases are mainly due to the PP biases, implying that NUBF effects are mainly due to unresolved variability of cloud inside the FOV, and to a lesser extent, to the horizontal photon transport which increases if

FOV increases. Finally, these results give an indication of the reliability of the lidar signals modelled using ICA approximation.

All these simulations and results are still a test bench to show the ability of the McRALI-FR simulation tool to study the impact of multiple scattering and 3D cloud radiative effects on remote sensing observations and products. Real detailed cloud case studies and statistical analysis of representative fine-structure 3-D cloud field effects on lidar and radar observables will be the topic of future papers.

**Code availability**

FORTRAN McRALI-FR codes can be obtained by contacting the corresponding author of this article.

**Author contributions**

FS, VS and GM: conceptual idea, supervision, methodology. AA and GM: numerical simulation and formal analysis. FS and

GM: funding acquisition. All of the authors contributed to developing of the McRALI code and to writing and revising the paper.

**Competing interests**

The authors declare that they have no conflict of interest.





**Appendix A**

| Acronym | Definition |
|---------|------------|
| A-train | Afternoon Constellation |
| ADM-Aeolus | Atmospheric Dynamics Mission |
| ALADIN | Atmospheric LAser Doppler Instrument |
| ATB | ATtenuated Backscatter |
| ATLID | ATmospheric LIDar |
| CALIPSO | Cloud-Aerosol Lidar and Infrared Pathfinder Satellite Observations |
| CNES | French National Centre for Space Studies |
| COD | Cloud Optical Depth |
| CPR | Cloud Profiling Radar |
| 3D | Three-Dimensional |
| 3DMCPOL | 3D POLarized Monte-Carlo atmospheric radiative transfer model |
| 3DMcPOLID | 3D Monte Carlo simulator of POLarized LIDar signals |
| DOMUS | DOppler MUltiple Scattering simulator |
| EECLAT | Expecting EarthCare, Learning from A-train |
| EarthCARE | Earth Clouds, Aerosol and Radiation Explorer mission |
| ECSIM | EarthCARE simulator |
| ESA | European Space Agency |
| FOV | Field Of View |
| ICA | Independent Column Approximation |
| INSU | French National Institute for Earth Sciences and Astronomy |



| HSR | High Spectral Resolution |
|---|---|
| MC | Monte Carlo |
| MS | Multiple Scattering |
| McRALI | Monte Carlo modeling of RAdar and LIdar signals |
| McRALI-FR | McRALI Frequency-Resolved simulator |
| MUSCLE | MUltiple SCattering in Lidar Experiments |
| NUBF | Non Uniform Beam Filling |
| PDF | Probability Density Function |
| PP | Plan-Parallel and homogenous cloud |
| RTE | Radiative Transfer Equation |
| SS | Single Scattering |

Table A1: List of acronyms used in this work and their definition.

**Appendix B**

Total, molecular and particulate $ATB$ are hereafter noted $ATB_t$, $ATB_m$ and $ATB_p$, respectively. According to Alkasem et al. (2017), PP bias can be understood from the following. Assuming null absorption and vertically constant atmospheric

5    properties, then $ATB(r)_t$ at position $r$ can be expressed as

$$ATB(r)_t = \left(P_m\sigma_m + P_p\sigma_p\right)e^{-2\int_0^r(\sigma_m+\gamma\sigma_p)dr} \tag{B1}$$

where $\sigma_m$ and $\sigma_p$ are the molecular and particulate scattering coefficients, respectively, $P_m$ and $P_p$ are the molecular and particulate phase functions at 180°, respectively, and $\gamma$ is a factor that takes into account the multiple scattering effects (Platt, 1973). In the same way, $ATB(r)_m$ and $ATB(r)_p$, by voluntarily omitting $r$, assuming that $\sigma_m$ and $\sigma_p$ are vertically constant, and introducing the thickness $\Delta r$ to lighten the writing, can be expressed as:

$$ATB_m = P_m\sigma_m e^{-2(\sigma_m+\gamma\sigma_p)\Delta r} \tag{B2}$$

$$ATB_p = P_p\sigma_p e^{-2(\sigma_m+\gamma\sigma_p)\Delta r} \tag{B3}$$





Let $\alpha$ the cloud coverage inside the lidar receiver FOV, then the molecular (i.e. $ATB_{m;IPA}$), the particulate (i.e. $ATB_{p;IPA}$) and

the total ATB (i.e. $ATB_{t;IPA}$) computed with the IPA cloud model can be written as:

$$ATB_{m;IPA} = (1-\alpha)P_m\sigma_m e^{-2\sigma_m\Delta r} + \alpha P_m\sigma_m e^{-2(\sigma_m+\gamma\sigma_p)\Delta r} \tag{B4.1}$$

$$ATB_{p;IPA} = \alpha P_p\sigma_p e^{-2(\sigma_m+\gamma\sigma_p)\Delta r} \tag{B4.2}$$

$$ATB_{t;IPA} = (1-\alpha)P_m\sigma_m e^{-2\sigma_m\Delta r} + \alpha(P_m\sigma_m + P_p\sigma_p)e^{-2(\sigma_m+\gamma\sigma_p)\Delta r} \tag{B4.3}$$

and the molecular (i.e. $ATB_{m;PP}$), the particulate (i.e. $ATB_{p;PP}$) and the total (i.e. $ATB_{t;PP}$) ATB computed with the PP cloud

model can be written as :

$$ATB_{m;PP} = P_m\sigma_m e^{-2(\sigma_m+\alpha\gamma\sigma_p)\Delta r} \tag{B5.1}$$

$$ATB_{p;PP} = \alpha P_p\sigma_p e^{-2(\sigma_m+\alpha\gamma\sigma_p)\Delta r} \tag{B5.2}$$

$$ATB_{t;PP} = (P_m\sigma_m + \alpha P_p\sigma_p)e^{-2(\sigma_m+\alpha\gamma\sigma_p)\Delta r} \tag{B5.3}$$

5   The PP molecular bias $\Delta ATB_{m;PP}$, estimated as:

$$\Delta ATB_{m;PP} = ATB_{m;PP} - ATB_{m;IPA} \tag{B6}$$

$$= P_m\sigma_m\left[e^{-2(\sigma_m+\gamma\alpha\sigma_p)\Delta r} - (1-\alpha)e^{-2\sigma_m\Delta r} - \alpha e^{-2(\sigma_m+\gamma\sigma_p)\Delta r}\right]$$

is always negative. The PP particulate bias $\Delta ATB_{p;PP}$, estimated as:

$$\Delta ATB_{p;PP} = ATB_{p;PP} - ATB_{p;IPA} = P_p\sigma_p\left\{\alpha\left[e^{-2(\sigma_m+\alpha\gamma\sigma_p)\Delta r} - e^{-2(\sigma_m+\gamma\sigma_p)\Delta r}\right]\right\} \tag{B7}$$

is always positive. Note that the smaller $\gamma$ is, the smaller $\Delta ATB_{p;PP}$. In other words, the larger multiple scattering effects are

the smaller the PP particulate bias. The PP total bias $\Delta ATB_{t;PP}$, estimated as:

$$\Delta ATB_{t;PP} = ATB_{t;PP} - ATB_{t;IPA} \tag{B8}$$

$$= (P_m\sigma_m + \alpha P_p\sigma_p)e^{-2\gamma(\sigma_m+\alpha\sigma_p)\Delta r} - (1-\alpha)P_m\sigma_m e^{-2\gamma\sigma_m\Delta r}$$

$$- \alpha(P_m\sigma_m + P_p\sigma_p)e^{-2\gamma(\sigma_m+\sigma_p)\Delta r}$$

is positive but becomes negative with increasing cloud optical depth, as explained in Alkasem et al. (2017).



**Acknowledgements**

This work is part of the French scientific community EECLAT project (Expecting EarthCare, Learning from A-train) (Luebke et al., 2018). The EECLAT community and research activities are supported by the National Center for Space Studies (CNES) and the National Institute for Earth Sciences and Astronomy (INSU). This work has also been in part supported by the

5   Programme National de Télédétection Spatiale (PNTS, http://www.insu.cnrs.fr/pnts), grant n°PNTS-2019-8.

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
