# Peer review of "McRALI: a Monte Carlo high spectral resolution lidar and Doppler radar simulator for three-dimensional cloudy atmosphere remote sensing"

_Atmospheric Measurement Techniques, 2020_

## Referee Comment (RC1) · Anonymous Referee #1 · 18 Sep 2020

The paper describes a MonteCarlo-based multiple scattering Doppler radar and lidar simulator. Though some of the described techniques replicate already published methods the work has its value because it offers a tool for both radars and lidars and represents an independent mean to evaluate results of other simulators developed by the scientific community. The simulator will have value especially with the launch of the EarthCARE mission. The scenarios proposed to demonstrate the capabilities are quite simplistic (e.g. only box clouds are used), but this is ok for presenting the potential applications of the tool.

[Figure]

I have few comments.

1) There are no results shown for polarization. It would be great to see some results for the other Stokes parameters (not simply the intensity)

2) I am a little bit surprised to see a mirror image from a Lambertian surface (but I may be radar-biased). Is this also present in lidar observations?

3) Line 16 in the intro: not true for the EC radar, no spectrum will be actually be provided.

4) line 14 page 4: "receiver shape" which shape??

5) Notation of Eq.8 is confusing, you are using first double and then single subscripts

6) Fig4, bottom left panels: not sure why in the SS result there are values below cloud base. There should be none.

7) Page 7: "The frequency shift due to satellite motion is deliberately ignored..." well it is not clear to me then how you can simulate the Doppler broadening due to the satellite motion itself.

8) Formula 19: the formula is not accounting for other causes of spectral widths (for sure microphysics should be accounted for!). Add missing terms.

Other minor comments: why not using the same color scheme for 65 and 650 micro-rad?

Line 9 in the introduction: not clear why this line is there, out of context.

Introduce titles for the Appendices

---

## Referee Comment (RC2) · Anonymous Referee #2 · 18 Sep 2020

Some time ago now, I reviewed an earlier version of this paper for another journal. Now, I find the work and presentation to be much improved ! I have only a few mainly minor comments/concerns.

Page 2: Around line 15:

Aeolus carries a Fizeau spectrometer, which measures the spectrum of the return +/- 0.33 pm around the emitted laser wavelength using 16 different frequency bins. Thus, Aeolus provides spectrally resolved data in the normal sense of the phrase. ATLID

separates the pure so-called "Mie" and Rayleigh backscatter returns. This is not the same as measuring the full spectrum. Please adjust the text accordingly.

Page 2: Line 20:

Simulation tools are steadily advancing thus allowing the exploration of the direct...way. Lidar and/or radar simulators are no exception.

Page 3: Line 5.

This text is not clear. Is DOMUS part of ECSIM or something separate ?

Section 2.2: The modelling of Doppler shifts is well described, however, the treatment of polarization is not described at all (despite the section title of " Modelling of idealized polarized backscattered power spectrum profiles") !  At least short description with references should be given.

Page 14: Lines 10-15.  The cross-talk coefficient used in this paper have been apparently derived assuming ideal behavior of the EarthCARE FP element.  In practice, the Airy function will be "blurred" due to the effects of non-ideal collimation of the beam, frequency jiter, surface roughness etc.. These factor all act to decrease the peak transmission and lower the FWHM. For a more realistic view on these parameters you should take into account the information in CEAS Space Journal (2019) 11:423–435 https://doi.org/10.1007/s12567-019-00284-6 (See Fig 9).

Looking at the FP characterization curve, it is certain that here will be substantially more Mie to Ray cross-talk than reflected by the present choice of coefficients. This fact will not alter any of the present papers conclusions (the increased X-talk will act mainly to reduce the SNR of the cross-talk corrected observations). Rather than redo the "EarthCARE" cases (which would be ideal but may require too-much time/effort) the authors could instead make it clear that the calculations shown are merely "EarthCARE like" but with an idealized modeled FP etc...

Page 14: Line 21 "(named by abuse of language)" ==> "the so-called"

Page 14: Line 26 "..step a simulated FP interferometer separates the ..."

Page 15: Figure 5 and associated text. It is likely worth pointing out the quasi-exponential decay of the below cloud molecular return towards single scattering return levels. This result is consistent with the cases shown by Donovan 2016.

Page 25: Lines 10-15: Have the the variance reduction techniques described by Buras et al. been employed in these calculations ? If so, it is work some more discussion regarding why these spikes in the spectrum remain. If not, then why were they not used ?
* * *

---

## Author Comment (AC1) · 20 Oct 2020

Response to Reviewer # 1

We thank the reviewer for his review and valuable comments. The manuscript has been modified according to the suggestions proposed by the reviewer. The remainder is devoted to the specific response item-by-item of the reviewer's comments.

*RC=Reviewer Comments*
AR=Author response
TC=Text Changes

**General Comments**

*The paper describes a MonteCarlo-based multiple scattering Doppler radar and lidar simulator. Though some of the described techniques replicate already published methods the work has its value because it offers a tool for both radars and lidars and represents an independent mean to evaluate results of other simulators developed by the scientific community. The simulator will have value especially with the launch of the EarthCARE mission. The scenarios proposed to demonstrate the capabilities are quite simplistic (e.g. only box clouds are used), but this is ok for presenting the potential applications of the tool.*

**Specific Comments**

1) *There are no results shown for polarization. It would be great to see some results for the other Stokes parameters (not simply the intensity)*

We used the McRALI code to simulate profiles of the volume depolarization ratio measured by the CALIOP/CALIPSO lidar in our previous work (see Figs. 7-8, Alkasem et al., 2017). In the same work, we demonstrated that our simulation of the linear and circular depolarization profiles in a C1 cloud are in good agreement with the published data (see Appendix A.3, Alkasem et al., 2017). The later result was confirmed by other authors that cited our work (see, e.g., Sato et al., 2019; Wang et al., 2019).
We prefer to not present our new results on polarization in this work because they would make the paper under review too long. Those results will be the subject of a separated publication.

We modified the last sentence of the revised paper as follows:

"Real detailed cloud case studies and statistical analysis of representative fine-structure 3-D cloud field effects on lidar and radar observables, while taking into account the polarization of the light, will be the topic of future papers."

2) *I am a little bit surprised to see a mirror image from a Lambertian surface (but I may be radar-biased). Is this also present in lidar observations?*

To our knowledge, there exist no publications showing a mirror image in lidar observations from the atmosphere. Indeed, signal to noise ratio (SNR) of lidar is generally much lower than the SNR of radar. Thus, it seems to be practically impossible to observe a mirror image with a spaceborne lidar, contrary to a spaceborne radar (Battaglia et al., 2010).
The results presented in figure 3 should be considered as a numerical and theoretical exercise that demonstrates the McRALI capacities. The simulations were performed with a very high number of photon trajectories, so that the numerical noise of the McRALI simulator is low. Under these idealized simulations conditions, we show that McRALI is capable not only to

simulate lidar / radar systems with inclined sighting, by taking into account the properties of the Lambertian surface, but also the mirror images (as it is seen in certain radar observations). It should be noted that to make the mirror image appear in this simulation, we have imposed a maximum surface albedo equal to 1.
We add these sentences at the end of section 2.4 :

"Signal to noise ratio (SNR) of lidars is generally much lower than the SNR of radars. Thus, in practice it is impossible to observe a mirror image with a spaceborne lidar, contrary to a spaceborne radar. Results presented in fig.3 should be considered as a numerical and theoretical exercise that demonstrates the McRALI capacities. The simulations were performed with a very high number of photon trajectories, so that the numerical noise of the McRALI simulator is very low. Under these idealized simulations conditions, we show that McRALI is able to simulate lidar / radar systems with inclined sighting by taking into account the properties of the Lambertian surface, but also the mirror images (as it is seen in certain radar observations). It should be noted that to make the mirror image appear in this simulation, we have imposed a maximum surface albedo equal to 1".

3) *Line 16 in the intro: not true for the EC radar, no spectrum will be actually be provided.*

We thank the reviewer for this comment. In the revised paper, and according to suggestions of the second reviewer, we have changed the sentence "The Atmospheric LAser Doppler INstrument (ALADIN) of the ADM-Aeolus, the ATmospheric LIDar (ATLID) and the Cloud Profiling Radar (CPR) of the EarthCARE mission will provide spectrally resolved data." by :

The Atmospheric LAser Doppler INstrument (ALADIN) of the ADM-Aeolus provides spectrally resolved data. Indeed, the Mie receiver is a Fizeau spectrometer combined with a charge-coupled detector that measures the spectrum of the return around the emitted laser wavelength using 16 different frequency bins (Stoffelen et al., 2005; Reitebuch, 2018). The ATmospheric LIDar (ATLID) signals of the EarthCARE mission will be optically filtered in such a way that the atmospheric Mie and Rayleigh scattering contributions are separated and independently measured (Pereira do Carmo et al., 2019).The radar echoes of the Cloud Profiling Radar (CPR) of the EarthCARE mission will be input to autocovariance analysis by means of the pulse-pair processing technique for the estimation of the Doppler properties (Zrnic, 1977 , Kollias et al., 2013; Kollias et al., 2018). Note however that ATLID and CPR will not provide the spectrally resolved data.

4) *line 14 page 4: "receiver shape" which shape??*

In the revised paper, we have changed "… and account for emitter and receiver shape" by :

"… and account for emitter and receiver patterns of the lidar (or radar) system".

*5) Notation of Eq.8 is confusing, you are using first double and then single subscripts*

We agree with the reviewer. In the submitted paper, before Eq.8, the unit director vector defined between the scatterer i and i + 1, is written $k_{i,i+1}$. Then, after Eq.8, this same vector is written $k_i$. In the revised paper, we have changed $\boldsymbol{k_{i,i+1}}$ by $\boldsymbol{\hat{k}_{i,i+1}}$ and we have added the definition of $\boldsymbol{k_i}$. in page 7, just after the Eq.8 :

"where $\boldsymbol{k_i}$ is the unit director vector defined between the scatterer i and i + 1".

*6) Fig4, bottom left panels: not sure why in the SS result there are values below cloud base. There should be none.*

In the submitted paper, the authors wanted to show in Fig4.c the values predicted by the theory using a vertical dotted line. But this way of doing things brings confusion since under the cloud, the theory does not foresee any value, as argued by the reviewer. In the revised paper we have removed these dotted lines and deleted the information in the legend and in the explanation of the figure.

*7) Page 7: "The frequency shift due to satellite motion is deliberately ignored..." well it is not clear to me then how you can simulate the Doppler broadening due to the satellite motion itself.*

In order to clarify how the Doppler broadening due to the satellite motion itself is simulated, we have modified this sentence (page 7) of the submitted paper "For example, at the second scattering event, the total frequency shift is computed as $\Delta f_{2;total} = \Delta f_1 + \Delta f'_2$, where $\Delta f'_2 = \frac{f_0}{c} \boldsymbol{v_2}. (\boldsymbol{k_1} - \boldsymbol{k'}_2)$, $\boldsymbol{k'}_2$ being the direction from the second scattering event to the detector (dotted blue line) which works with the local estimate method." by :

For example, at the second scattering event, the total frequency shift is computed as $\Delta f_{2;total} = \Delta f_1 + \Delta f'_2 + \Delta f'_{sat}$, where $\Delta f'_2 = \frac{f_0}{c} \boldsymbol{v_2}. (\boldsymbol{k_1} - \boldsymbol{k'}_2)$, $\boldsymbol{k'}_2$ being the direction from the second scattering event to the detector (dotted blue line) which works with the local estimate method and where $\Delta f'_{sat} = -\frac{f_0}{c} \boldsymbol{v_{sat}}. (\boldsymbol{k_0} - \boldsymbol{k'}_2)$, $\boldsymbol{v_{sat}}$ being the satellite velocity.

*8) Formula 19: the formula is not accounting for other causes of spectral widths (for sure microphysics should be accounted for!). Add missing terms.*

In page 12 of the revised paper, we have changed "On Fig. 4c the MS and SS Doppler velocity spectral width profiles are drawn.Under SS approximation, the Doppler velocity spectral width is given by (Tanelli et al., 2002):

$$\sigma_{Dop}^2 = \sigma_{turb}^2 + \left( \frac{\rho_R v_{sat}}{2\sqrt{\ln(2)}} \right)^2 \tag{19}$$

where $v_{sat}$ is the satellite velocity relative to the ground and $\rho_R$ is the Gaussian (3-dB) FOV half-angle" by :

On Fig. 4c the MS and SS Doppler velocity spectral width profiles are drawn. Under SS approximation, the Doppler velocity spectral width $\sigma_{Dop}$ is given by (Kobayashi et al., 2003 ; Battaglia et al., 2013) $\sigma_{Dop}^2 = \sigma_{hydro}^2 + \sigma_{shear}^2 + \sigma_{turb}^2 + \sigma_{motion}^2$, where $\sigma_{hydro}$ is due to the spread of the terminal fall velocities of hydrometeors of different size, $\sigma_{shear}$ is the broadening due to the vertical shear of vertical wind, $\sigma_{turb}$ is the broadening of the vertical wind due to turbulent motions in the atmosphere and $\sigma_{motion}$ is the spread caused by the coupling between the platform motion and the vertical wind shears of the horizontal winds. For a Gaussian circular antenna pattern, assuming zero fall velocities of hydrometeors and no wind shear, $\sigma_{Dop}$ is given by (Tanelli et al., 2002):

$$\sigma_{Dop}^2 = \sigma_{turb}^2 + \left(\frac{\rho_R v_{sat}}{2\sqrt{\ln(2)}}\right)^2 \qquad (19)$$

where $v_{sat}$ is the satellite velocity relative to the ground and $\rho_R$ is the Gaussian (3-dB) FOV half-angle".

9) *Other minor comments: why not using the same color scheme for 65 and 650 microrad?*

In the revised paper, same colors are used for 65 and 650 microrad.

10) *Line 9 in the introduction: not clear why this line is there, out of context.*

In the revised paper, the sentence "CALIPSO and CloudSat missions were then extended for other 3 years (see, e.g., Vandemark et al., 2017)" is deleted.

11) *Introduce titles for the Appendices*

We have added titles in the revised paper.
Title of appendix A is "Definition of acronyms"
Title of appendix B is "Estimation of the PP bias of molecular, particulate and total ATB as a function of cloud coverage and multiple scattering intensity for the box cloud model"

References

K. Sato, H. Okamoto, and H. Ishimoto, "Modeling the depolarization of space-borne lidar signals," Opt. Express 27, A117-A132 (2019) https://doi.org/10.1364/OE.27.00A117

Z. Wang, S. Cui, Z. Zhang, J. Yang, H. Gao, F. Zhang, Theoretical extension of universal forward and backward Monte Carlo radiative transfer modeling for passive and active polarization observation simulations, Journal of Quantitative Spectroscopy and Radiative Transfer, 235, 2019, p. 81-94, https://doi.org/10.1016/j.jqsrt.2019.06.025.

---

## Author Comment (AC2) · 20 Oct 2020

Response to Reviewer # 2

We thank the reviewer for his review and valuable comments. The manuscript has been modified according to the suggestions proposed by the reviewer. The remainder is devoted to the specific response item-by-item of the reviewer's comments.

*RC=Reviewer Comments*
AR=Author response
TC=Text Changes

**General Comments**

*Some time ago now, I reviewed an earlier version of this paper for another journal. Now, I find the work and presentation to be much improved! I have only a few mainly minor comments/concerns.*

We thank the referee for all comments on the early version of this paper, which allowed us to progress and improve the content of this new version.

**Specific Comments**

*1) Aeolus carries a Fizeau spectrometer, which measures the spectrum of the return +/- 0.33 pm around the emitted laser wavelength using 16 different frequency bins. Thus, Aeolus provides spectrally resolved data in the normal sense of the phrase. ATLID separates the pure so-called "Mie" and Rayleigh backscatter returns. This is not the same as measuring the full spectrum. Please adjust the text accordingly.*

We thank the reviewer for this comment. In the revised paper, and according to suggestions of the first reviewer, we have changed the sentence "The Atmospheric LAser Doppler INstrument (ALADIN) of the ADM-Aeolus, the ATmospheric LIDar (ATLID) and the Cloud Profiling Radar (CPR) of the EarthCARE mission will provide spectrally resolved data." by :

The Atmospheric LAser Doppler INstrument (ALADIN) of the ADM-Aeolus provides spectrally resolved data. Indeed, the Mie receiver is a Fizeau spectrometer combined with a charge-coupled detector that measures the spectrum of the return around the emitted laser wavelength using 16 different frequency bins (Stoffelen et al., 2005; Reitebuch, 2018). The ATmospheric LIDar (ATLID) signals of the EarthCARE mission will be optically filtered in such a way that the atmospheric Mie and Rayleigh scattering contributions are separated and independently measured (Pereira do Carmo et al., 2019).The radar echoes of the Cloud Profiling Radar (CPR) of the EarthCARE mission will be input to autocovariance analysis by means of the pulse-pair processing technique for the estimation of the Doppler properties (Zrnic, 1977 , Kollias et al., 2013;  Kollias et al., 2018). Note however that ATLID and CPR will not provide the spectrally resolved data.

*2) Page 2: Line 20: Simulation tools are steadily advancing thus allowing the exploration of the direct...way. Lidar and/or radar simulators are no exception.*

In the revised paper, we have changed "Simulation tools are steadily advancing thus allowing the exploration of the direct...way. Lidar and/or radar simulators are no exception" by

"Lidar and/or radar simulators are steadily advancing hence allowing to explore direct and inverse problems in a cost-effective way".

*3) Page 3: Line 5. This text is not clear. Is DOMUS part of ECSIM or something separate ?*

DOMUS is not a part of ECSIM. In the revised paper, we have added in page 3 :

"Note that DOMUS is not a part of ECSIM."

*4) Section 2.2: The modelling of Doppler shifts is well described, however, the treatment of polarization is not described at all (despite the section title of " Modelling of idealized polarized backscattered power spectrum profiles") ! At least short description with references should be given.*

This remark of reviewer 2 joins the comment of reviewer 1 which asks for results on polarization.
We used the McRALI code to simulate profiles of the volume depolarization ratio measured by the CALIOP/CALIPSO lidar in our previous work (see Figs. 7-8, Alkasem et al., 2017). In the same work, we demonstrated that our simulation of the linear and circular depolarization profiles in a C1 cloud are in good agreement with the published data (see Appendix A.3, Alkasem et al., 2017). The later result was confirmed by other authors that cited our work (see, e.g., Sato et al, 2019; Wang et al, 2019).
We prefer not to present our new results on polarization in this work because they would make the paper under reviewing too long. Those results will be a subject of a separated publication.

We modified the last sentence of the revised paper as following:
"Real detailed cloud case studies and statistical analysis of representative fine-structure 3-D cloud field effects on lidar and radar observables, while taking into account the polarization of the light, will be the topic of future papers."
Moreover, in the revised paper, we have modified the title of section 2.2 by deleting the word "polarized".

*5) Page 14: Lines 10-15. The cross-talk coefficient used in this paper have been apparently derived assuming ideal behavior of the EarthCARE FP element. In practice, the Airy function will be "blurred" due to the effects of non-ideal collimation of the beam, frequency jiter, surface roughness etc.. These factor all act to decrease the peak transmission and lower the FWHM. For a more realistic view on these parameters you should take into account the information in CEAS Space Journal (2019) 11:423–435 https://doi.org/10.1007/s12567-019-00284-6 (See Fig 9).*

*Looking at the FP characterization curve, it is certain that here will be substantially more Mie to Ray cross-talk than reflected by the present choice of coefficients. This fact will not alter any of the present papers conclusions (the increased X-talk will act mainly to reduce the SNR of the*

*cross-talk corrected observations). Rather than redo the "EarthCARE" cases (which would be ideal but may require too-much time/effort) the authors could instead make it clear that the calculations shown are merely "EarthCARE like" but with an idealized modeled FP etc...*

We thank the reviewer for the information, comments (and the publication reference) on cross-talk effects in ATLID which are not all taken into account in the McRALI simulator, the one using an idealized modeled FP interferometer. We also agree that this fact does not alter any of our conclusions. In order to clarify the fact that our calculations are carried out under idealized conditions (idealized FP interferometer, as suggested by the reviewer), we have added in the revised paper, page 15, this sentence:

Note that the cross-talk coefficients used in this paper assume ideal behavior of the ATLID FP interferometer. In practice, the Airy function will be "blurred" due to the effects of non-ideal collimation of the beam, frequency jiter, surface roughness and so on. All these factors end to decrease the peak transmission and lower the full width at half maximum (see the Fig. 9 in Pereira do Carmo et al., 2019). It is important to keep in mind that all the calculations shown in this paper are merely "EarthCARE like" but with an idealized modeled FP interferometer.

*Page 14: Line 21 "(named by abuse of language)" ==> "the so-called"*

Corrected in the revised paper.

*Page 14: Line 26 "..step a simulated FP interferometer separates the ..."*

Corrected in the revised paper.

*Page 15: Figure 5 and associated text. It is likely worth pointing out the quasi exponential decay of the below cloud molecular return towards single scattering return levels. This result is consistent with the cases shown by Donovan 2016.*

We thank the reviewer for this remark. We have added these sentences in the revised paper (page 16, line 5-6 in the revised manuscript) :

It is likely worth pointing out the quasi exponential decay of the below cloud molecular return towards single scattering return levels. This result is consistent with the cases shown by Donovan (2016).

*Page 25: Lines 10-15: Have the the variance reduction techniques described by Buras et al. been employed in these calculations ? If so, it is work some more discussion regarding why these spikes in the spectrum remain. If not, then why were they not used ?*

The variance reduction techniques described by Buras et al. has not been employed in calculations of this work.
With reference to the variance reduction techniques (VRTs), the Monte-Carlo code McRALI has two separated sets of subroutines. With the first set (without VRTs), simulations are done only using the local estimate method. In the second set (with VRTs), the methods and equations of the work by Buras and Mayer (2011), hereafter BM2011, are implemented. A McRALI user

can choose to do simulations with or without the VRTs. The user must assign the set of parameters (see, Section 2.6 of BM20110) when simulations with the VRTs are performed.

We consider a set of VRTs parameters to be acceptable when there is good amelioration in the computing time and there are no biases between simulations with and without VRTs. For instance, our simulations of the MUSCLE cases are in very good agreement with the MUSCLE community results (see, Section A.1, Alkasem et al. 2017). The difference between the ratios of multiple-to-single scattering, which were computed using the McRALI code with and without VRTs, is within ±5%. The VRTs computing time is 100 times faster. The important point of the MUSCLE cases is the extinction coefficient value that is of 17,25 km-1.

When the extinction coefficient value is rather low (1 km-1 or lower), it is especially difficult and time consuming to get an acceptable set of VRTs parameters. Unfortunately, we have not succeeded to find a unique set of VRTs parameters, which is acceptable for different lidar types and configurations, and different particles phase functions. Thus, we decided to perform simulations of that work without the VRTs.
Of course, the statement above has to be considered as only our personal experience. Thorough investigations are needed before it will be accepted or rejected.

We added to the revised manuscript (page 3 line 25) the following text.

"All simulations of this work were done without application of the variance reduction techniques."

References

K. Sato, H. Okamoto, and H. Ishimoto, "Modeling the depolarization of space-borne lidar signals," Opt. Express 27, A117-A132 (2019) https://doi.org/10.1364/OE.27.00A117

Z. Wang, S. Cui, Z. Zhang, J. Yang, H. Gao, F. Zhang, Theoretical extension of universal forward and backward Monte Carlo radiative transfer modeling for passive and active polarization observation simulations, Journal of Quantitative Spectroscopy and Radiative Transfer, 235, 2019, p. 81-94, https://doi.org/10.1016/j.jqsrt.2019.06.025.